# Demystifying the Role of Histone Demethylases in Colorectal Cancer: Mechanisms and Therapeutic Opportunities

**DOI:** 10.3390/cimb47040267

**Published:** 2025-04-09

**Authors:** Yuanbin Liu, Min Huang, Xia Tian, Xiaodong Huang

**Affiliations:** Department of Gastroenterology, Tongren Hospital of Wuhan University (Wuhan Third Hospital), Wuhan 430060, China; 2016302180347@whu.edu.cn (Y.L.); 15927069508@163.com (M.H.)

**Keywords:** histone demethylases, colorectal cancer, epigenetics, histone demethylation, lysine demethylases

## Abstract

Histone demethylases (HDMs) play a pivotal role in colorectal cancer (CRC) progression through dynamic epigenetic regulation. This review summarizes the role and therapeutic potential of HDM in CRC. HDMs primarily target lysine (K) for demethylation (lysine demethylase, KDM). The KDM family is divided into the lysine-specific demethylase family and the Jumonji C domain-containing family. HDMs play complex roles in CRC cell proliferation, invasion, migration, stemness, epithelial–mesenchymal transition, immune response, and chemoresistance through epigenetic regulation of different histone demethylation sites. Increasing evidence suggests that KDM may interact with certain factors and regulate CRC tumorigenesis by modulating multiple signaling pathways and affecting the transcription of target genes. These processes may be regulated by upstream genes and thus form a complex epigenetic regulatory network. However, the potential roles and regulatory mechanisms of some HDMs in CRC remain understudied. Preclinical studies have revealed that small-molecule inhibitors targeting HDM impact the activity of specific genes and pathways by inhibiting specific HDM expression, thereby reshaping the tumorigenic landscape of CRC. However, the clinical translational potential of these inhibitors remains unexplored. In conclusion, HDMs play a complex and critical role in CRC progression by dynamically regulating histone methylation patterns. These HDMs shape the malignant behavior of CRC by influencing the activity of key pathways and target genes through epigenetic reprogramming. Targeting HDM may be a promising direction for CRC treatment. Further exploration of the role of specific HDMs in CRC and the therapeutic potential of HDM-specific inhibitors is needed in the future.

## 1. Introduction

Colorectal cancer (CRC) is one of the most common malignant tumors worldwide and poses a major threat to human health. According to global cancer statistics, the number of new cases of CRC in 2022 was estimated to be more than 1.9 million, accounting for 9.6% of all cancers, and ranking third in the global cancer incidence rate [1]. The number of CRC-related deaths was approximately 904,000 cases, accounting for 9.3% of all cancer deaths and ranking second in the global cancer mortality rate [1]. In recent years, the incidence of CRC in China has shown a significant upward trend, and China currently ranks first in the world in terms of new CRC cases and CRC-related deaths [2,3]. Despite advances in surgical techniques, chemotherapy, and targeted therapies, the overall survival of CRC patients remains unsatisfactory, highlighting the need for a deeper understanding of the molecular mechanisms of CRC and the identification of new therapeutic targets [4,5].

The onset and progression of CRC is a complex process involving the interaction of genetic mutations, epigenetic alterations, and the tumor microenvironment [6,7,8,9]. In recent years, the role of epigenetic regulation, including DNA methylation, histone modifications, and noncoding RNAs, in the development of CRC has received much attention [10,11,12]. Unlike genetic mutations, epigenetic alterations dynamically regulate gene expression through reversible chromatin remodeling and are closely associated with tumor microenvironment remodeling, stem cell property maintenance and treatment resistance [13,14,15,16]. Histone modification is one of the core mechanisms of epigenetic regulation and plays a key role in the regulation of gene expression and chromatin structure. Histone demethylation is able to activate or repress gene expression through precise regulation of chromatin opening state and gene transcriptional activity by histone demethylases (HDMs), a class of enzymes capable of removing methyl groups from specific amino acid (mainly lysine [K]) residues on histones, which have emerged as key factors in the regulation of gene expression and cellular function [17,18,19,20]. Numerous studies have demonstrated that a variety of HDMs are abnormally highly expressed in CRC and drive the malignant phenotype of tumors through epigenetic reprogramming, whereas some of them exhibit tumor suppressor functions under specific conditions [21,22,23,24]. Increasing evidence suggests that HDMs are involved in the pathogenesis of CRC and have been shown to govern various biological processes, including, but not limited to, cell proliferation, apoptosis, chemoresistance, and immune responses [25,26,27,28]. Several small-molecule inhibitors and natural compounds targeting HDM have shown potential to inhibit CRC growth and metastasis in preclinical models, although research remains sparse [21,29,30]. In addition, targeting HDMs in combination with immune checkpoint inhibitors can synergistically remodel the immune microenvironment and enhance anti-tumor effects [31]. Accordingly, these studies demonstrate that HDMs are key players in CRC occurrence and progression, and their diverse roles in regulating gene expression and cellular function highlight their potential as therapeutic targets.

However, there is no comprehensive review summarizing current knowledge and recent advances in the mechanisms of action and therapeutic potential of HDM in CRC. This review aims to comprehensively summarize the current knowledge on HDMs and provide insights into their roles in CRC, offering a foundation for future research and therapeutic development. Key questions include how specific HDMs dynamically regulate histone methylation patterns to drive CRC malignancy, what are the upstream regulators and downstream effectors of HDMs in the CRC signaling pathway, and whether targeting HDMs can overcome drug resistance or immune evasion in CRC. HDMs are emerging as key epigenetic regulators of CRC that modulate tumor cell plasticity, stemness, and microenvironmental interactions. Importantly, HDM dysregulation is associated with chemotherapy resistance, making it a promising therapeutic target. This review provides a comprehensive overview of current knowledge on HDMs in CRC, including classification and composition of HDMs, elucidation of their roles in CRC hallmarks, dissection of the interactions between HDMs and signaling pathways, critical evaluation of preclinical HDM inhibitors, and future research directions.

## 2. Overview of CRC Pathogenesis and Signaling Pathways

The occurrence and development of CRC is a complex process involving the abnormal activation or inhibition of multiple intra- and extracellular signaling pathways. In recent years, with the rapid development of molecular biology and multi-omics technology, the understanding of the pathogenesis of CRC and related signaling pathways has been continuously advanced. The pathogenesis of CRC is a process driven primarily by genetic mutations, epigenetic alterations, microenvironmental dysregulation, gut microbiota dysbiosis, and immune escape, which can progress from the adenoma–carcinoma sequence, the serrated pathway, and the inflammatory pathway [32,33,34,35]. The development of CRC is closely associated with chromosomal instability and mutations in a variety of genes (e.g., *APC*, *KRAS*, *TP53*, and *SMAD4*), which can lead to abnormalities in processes such as cell proliferation, apoptosis, differentiation, and migration [9,36]; 15% of CRC cases exhibit microsatellite instability (MSI), caused primarily by mutations in DNA mismatch repair (MMR) genes or epigenetic silencing [37]. Epigenetic alterations also play an important role in the development and progression of CRC, including DNA methylation (hypermethylation of CpG islands), histone modifications (including methylation, acetylation, phosphorylation, etc.), and noncoding RNAs (e.g., miRNAs, lncRNAs) [12]. The tumor microenvironment (TME), which mainly includes tumor-associated fibroblasts, immune cells, extracellular matrix, and signaling molecules, plays a key role in key events of CRC progression such as angiogenesis, immune evasion, metabolic reprogramming, chemoresistance, and metastasis [6,35,38,39]. Recent studies have highlighted that gut microbiota dysbiosis is an important pathogenetic mechanism of CRC and constitutes an important component of TME, with potential translational and clinical implications in the diagnosis, prevention and treatment of CRC [32,40,41].

Dysregulation of signaling cascade responses has an important mediating role in CRC pathogenesis, including, but not limited to, the Wnt/β-catenin signaling pathway, RAS/MAPK signaling pathway, PI3K/AKT/mTOR signaling pathway, JAK/STAT signaling pathway, TGF-β/Smad signaling pathway, Notch signaling pathway, Hippo signaling pathway, and Hedgehog signaling pathway [33,42,43]. These pathways have complex interactions, and aberrant activation of these complex regulatory networks plays a critical role in CRC occurrence and development and provides targets and strategies for precision therapy of CRC. The Wnt signaling pathway is one of the key signaling pathways regulating CRC cell proliferation, differentiation, apoptosis, autophagy, stem cell properties, and tumor microenvironment remodeling, and aberrant activation of the Wnt signaling pathway is present in approximately 90% of CRC cases [44,45]. The MAPK signaling pathway mainly includes the RAS-RAF-MEK-ERK cascade, which regulates cell proliferation, differentiation, survival, apoptosis, etc. [46]. Overactivation of the MAPK pathway (usually caused by *KRAS* or *BRAF* mutations) is also involved in CRC progression by promoting angiogenesis, invasion, and metastasis [47]. In addition, the MAPK pathway cross-talks with other signaling pathways (e.g., PI3K/AKT, Wnt/β-catenin), which together promote CRC pathogenesis and development [48]. The PI3K/AKT/mTOR signaling pathway plays a key role in the regulation of cell proliferation, growth, apoptosis, survival, and migration and is implicated in metabolic reprogramming and therapeutic resistance in CRC [49]. Abnormal activation of the PI3K/AKT pathway in CRC is often caused by mutations in genes encoding key components of the signaling pathway (e.g., PIK3CA), PTEN deletion, or RTK overexpression [50]. In CRC, aberrant activation of the JAK/STAT pathway has been extensively reported to profoundly affect key biological processes such as cell proliferation, survival, migration, invasion, cancer stem cell (CSC) maintenance and immune evasion [51]. The TGF-β/Smad signaling pathway has a dual role in CRC. In normal cells and early stages of CRC, the TGF-β/Smad pathway mainly functions as a tumor suppressor. However, in the advanced stages of CRC progression, the TGF-β/Smad pathway undergoes a functional shift to become a pro-tumorigenic factor [52]. This transition is usually associated with mutations or deletions in TGF-β receptors (e.g., TGFβRII) or Smad4, leading to tumor cell migration and invasion, dysregulation of TME, and promotion of tumor immune escape and angiogenesis [52,53]. The activation of oncogenic Notch signaling pathway promotes proliferation and inhibits apoptosis of CRC cells [54]. In addition, the Notch pathway plays a key role in maintaining the self-renewal and differentiation of intestinal stem cells, and its aberrant activation can lead to excessive proliferation of stem cells and promote tumorigenesis [55]. Notch signaling also plays an important role in inducing epithelial–mesenchymal transition (EMT) and promoting tumor angiogenesis [56]. In CRC, the Hippo pathway (YAP and TAZ are the main effector molecules) is closely associated with tumorigenesis, progression and metastasis and can interact with other signaling pathways to promote tumor cell proliferation and invasion [57]. In addition, the activation of YAP/TAZ promotes TME dysregulation and inhibits anti-tumor immune responses [57]. The role of the Hedgehog pathway in CRC remains controversial [58,59]. However, most studies have shown that Hedgehog pathway activation enhances tumor cell proliferation and survival [58]. In addition, the Hedgehog pathway may promote CRC progression by maintaining CRC stem cell self-renewal, inducing EMT, promoting angiogenesis and facilitating immune escape, and modulating the tumor microenvironment [58]. In addition, AMPK signaling pathway, VEGF/VEGFR signaling pathway, HGF/cMET signaling pathway, immune checkpoint signaling pathway, RHOA signaling pathway, ErbB signaling pathway, Nrf2/Keap1 signaling pathway and NF-κB signaling pathway may also be involved in the tumorigenesis and progression of CRC [33,60].

## 3. Overview of Mechanisms of Histone Methylation, HDM Classification and Composition

In eukaryotic cells, DNA exists in the form of chromatin, and nucleosomes are the basic building blocks of chromatin. The core histones of the nucleosome (including histones H2A, H2B, H3, H4) have two structural domains: the histone folding domain and the amino-terminal domain. The amino-terminal domain is located outside the nucleosome core like a “histone tail” and is rich in amino acid residues that can be covalently modified [61]. Histone methylation is a key epigenetic modification that dynamically regulates gene expression by altering chromatin structure. This process involves the addition of methyl groups to lysine or arginine residues in histone tails and is mediated by histone methyltransferases (HMTs) [62]. Methylation can activate or repress transcription depending on the specific residue and degree of methylation (e.g., mono-, di-, or trimethylation). HDMs promote the reversibility of these modifications, allowing cells to rapidly adapt to environmental and developmental cues [62]. HDMs remove methyl groups through an oxidation reaction-dependent catalytic mechanism and require specific cofactors (e.g., flavin adenine dinucleotide and α-ketoglutarate) that play a key role in the catalytic process [63]. Histone demethylation affects the binding of transcription factors and other chromatin-associated proteins by altering the open or closed state of chromatin.

A schematic overview of the mechanism of HDMs in CRC is presented in Figure 1. The vast majority of HDMs are lysine demethylases (KDMs). KDMs are divided into two main groups based on their catalytic mechanisms and structural domains: the lysine-specific demethylase (LSD) family and the Jumonji C (JmjC) domain-containing (JMJD) family [64,65]. The LSD family mainly removes mono- and dimethyl groups on lysine residues (me1 and me2), whereas the JMJD family removes mono-, di- and trimethyl groups (me1, me2, and me3) on lysine residues [17,19]. The LSD family contains a flavin adenine dinucleotide-dependent amine oxidase structural domain and relies on flavin adenine dinucleotide to catalyze the demethylation of mono/demethylated lysines, whereas the JMJD family contains the JmjC structural domain and removes methylation modifications via Fe^2+^ and α-ketoglutarate-dependent oxidative reactions [66,67]. In general, H3K4, H3K36, and H3K79 methyl markers are transcriptionally active markers, whereas H3K9, H3K27, and H4K20 methylation is considered to silence gene expression [17]. However, these methylation marks may also play opposite roles under certain circumstances, and their specific functions depend on the degree of methylation, genomic location, and synergistic effects with other modifications. These enzymes regulate gene activation or repression by targeting specific histone lysine sites, which in turn affects cell proliferation, differentiation, and migration [66,68]. Histone arginine demethylation has been studied relatively little, and, to date, there are no definitive reports on specific arginine demethylases [64]. To date, two putative histone arginine demethylases, PAD4 (peptidyl arginine deiminase 4) and JMJD6, have been identified [69,70,71]. However, the demethylation function of both enzymes has also been questioned by studies [64]. A study has shown that the primary function of JMJD6 is to catalyze lysyl hydroxylation of the RNA splicing-related protein U2AF65 [72]. It has been suggested that PAD4 is not a histone arginine demethylase because it acts on both methylated and unmethylated arginine [64,73]. More recent evidence suggests that all human KDM5s, in addition to demethylating lysine, also possess N-methyl arginine demethylation functions [74]. HDM may be involved in CRC progression through demethylation-independent manners. As the aim was to summarize the mechanisms by which HDM regulates CRC through demethylation, we did not include these articles in this review. Therefore, since JMJD6 is involved in CRC progression through HDM-independent manners in a few publications [75,76], we will not discuss it here.

The LSD family contains only two KDMs, KDM1A and KDM1B. According to a previous high-quality review, there are currently 31 human proteins containing the JmjC structural domain, and 17 have been validated to have HDM functions [77]. They are KDM2A, KDM2B, KDM3A, KDM3B, KDM4A, KDM4B, KDM4C, KDM4D, KDM5A, KDM5B, KDM5C, KDM5D, KDM6A, KDM6B, KDM7A, KDM7B, and NO66 [77]. Several KDMs were previously considered to have controversial or unproven demethylation functions, including KDM3C, KDM4E, KDM6C, KDM7C, KDM8, and RIOX2, but were considered as KDMs in more recent research (although controversies remained in some) [78,79]. HDM classification, composition, and substrates are shown in Figure 2. The names, aliases, and histone substrates of these HDMs (both validated and potentially controversial) are summarized in Table 1.

## 4. LSD Family and CRC

### 4.1. KDM1A and CRC

The role of KDM1A in CRC has been extensively investigated (Table 2). Human KDM1A gene deletion is associated with advanced CRC staging [80]. Hayami et al. first demonstrated that KDM1A was significantly upregulated in CRC samples from Japanese patients [81]. It was shown that KDM1A is a regulator of DNA methyltransferase I stability [82]. Jin et al. found that KDM1A deletion (KDM1A heterozygous and homozygous knockout) in the HCT116 cell line significantly inhibited CRC cell proliferation in vitro and in vivo but did not lead to changes in H3K4me2 and H3K9me2 at the global cellular level and did not affect the protein levels or function of p53 or DNA methyltransferase I [83]. Using immunohistochemical staining, Jie et al. found that KDM1A was overexpressed in colon cancer tissues and significantly correlated with tumor TNM stage, lymph node and distant metastasis, and poorer prognosis [84]. In addition, KDM1A expression was negatively correlated with E-cadherin expression [84]. Consistently, Ding et al. revealed that KDM1A was able to bind to the E-cadherin gene promoter and reduce H3K4me2 levels, thereby promoting EMT [85]. In vitro pharmacological inhibition of KDM1A activity or siRNA knockdown of its expression significantly suppressed proliferation and invasion and induced apoptosis in colon cancer cells, suggesting that targeted inhibition of KDM1A may provide a new direction for the treatment of colon cancer [85]. Huang et al. found that KDM1A activated the Wnt/β-catenin signaling pathway through downregulation of DKK1, an antagonist of the Wnt/β-catenin signaling pathway, whereas in vitro KDM1A knockout inhibited tumorigenicity [86]. KDM1A is important for CSC stemness characterization maintenance in CRC [87]. Hsu et al. demonstrated that treatment with the KDM1A inhibitor CBB1003 could significantly inhibit the proliferation and clone-forming ability of CRC cells through inhibition of LGR5 (leucine-rich repeat-containing G-protein-coupled receptor 5), a CRC stem cell marker and Wnt signaling target, suggesting that KDM1A may affect the growth of CRC cells and the maintenance of CSCs through the modulation of key molecules in the Wnt/β-catenin signaling pathway [88]. A recent study demonstrated that KDM1A promoted tumor progression in CRC through multiple mechanisms, including maintenance of tumor stem cell properties, promotion of cell proliferation and migration, inhibition of apoptosis, and involvement in DNA damage response [89]. Transcriptomic and proteomic analyses revealed that KDM1A silencing induced upregulation of the expression of genes associated with cell differentiation while downregulating genes associated with the maintenance of stem cell properties [89]. In addition, KDM1A silencing was found to lead to cytoskeletal remodeling, metabolic reprogramming, altered mitochondrial function, and upregulation of the expression of miR-506-3p, which is known to have oncogenic effects [89].

Ding et al. showed that the long noncoding RNA (lnc RNA) HOXA-AS2 could bind to KDM1A and EZH2 (enhancer of zeste homolog 2, an HMT) and recruit them to the promoter regions of p21 (a cell cycle inhibitor) and Krüppel-like factor 2 (a tumor suppressor), thereby repressing the transcription of these genes [90], while Tian et al. showed that another lncRNA, FEZF1-AS1, also modulated CRC cell growth by interacting with KDM1A and inhibiting Krüppel-like factor 2 expression [91]. Another pseudogene-derived lncRNA, DUXAP10, could also bind to KDM1A to promote CRC cell growth and reduce apoptosis by silencing the expression of p21 and PTEN [92]. Liu et al. observed that lncRNA LINC00586 recruited KDM1A to the ASXL1 (a tumor suppressor) promoter region and epigenetically silenced ASXL1 expression, and that INC00586 knockdown inhibited HCT116 and LoVo cell viability, migration, and EMT as well as tumorigenesis in nude mice xenografted with HCT116 cells [93].

Hong et al. demonstrated that RIOK1 promoted CRC tumor growth and metastasis in vitro and in vivo, whereas KDM1A could increase the stability of RIOK1 by demethylating it (K411 site) [94]. Ding et al. found that miR-137-3p was poorly expressed in CRC, whereas inhibition of miR-137-3p upregulated KDM1A expression and promoted CRC cell invasiveness, and genetic or pharmacological inhibition of KDM1A reduced hypoxia-induced CRC cell migration and invasion by suppressing EMT [95]. Li et al. reported that ZY0511, a KDM1A-specific inhibitor, exerts antiproliferative effects on CRC cells by upregulating the expression of DNA Damage Inducible Transcript 4, a known mTORC1 inhibitor, through altering the H3K4 methylation level at the promoter [96]. Another study revealed that KDM1A was a potential marker of 5-fluorouracil resistance in CRC, and ZY0511 synergized with 5-fluorouracil in vitro to reduce the viability and migration of CRC cells to enhance the anticancer effects of 5-fluorouracil [97]. Interestingly, ZY0511 also profoundly altered lipid metabolism (e.g., sphingolipid and glycerophospholipid disorders) in human CRC cells, which may be associated with cancer progression [98].

Zhang et al. demonstrated that KDM1A upregulated tetraspanin 8 expression to promote CRC cell proliferation and migration by decreasing H3K9me2 occupancy at the tetraspanin 8 promoter, whereas tetraspanin 8 promoted EMT in a KDM1A-dependent manner [99]. Miller et al. demonstrated that KDM1A was a key regulator of intestinal enteroendocrine cell specification and that KDM1A deficiency impaired the formation of enteroendocrine progenitor cells and reduced CRC tumor growth and metastasis [100]. A more recent study demonstrated that KDM1A and CoREST2 mediate STAT3 demethylation and enhance STAT3 chromatin binding to promote enteroendocrine cell specification in mucinous CRC [101]. The knockdown of CoREST2 in an in situ xenograft mouse model resulted in reduced primary tumor growth and lung metastasis, suggesting that disruption of the interaction between KDM1A, CoREST2, and STAT3 serves as a therapeutic strategy to target neuroendocrine differentiation [101]. Soldani et al. showed that the metabolic axis between riboflavin and KDM1A may be involved in regulating the morphology of tumor-associated macrophages in patients with colorectal liver metastases, particularly in the inflammatory microenvironment, suggesting that KDM1A may be involved in TME regulation in CRC patients [102].

### 4.2. KDM1B and CRC

The role of KDM1B in colorectal cancer (CRC) remains poorly characterized (Table 2). Cai et al. found that the expression of KDM1B in CRC tissues was significantly higher than that in normal tissues, and overexpression of KDM1B promoted the proliferation of CRC cells and inhibited apoptosis [25]. Mechanistic studies showed that KDM1B promoted CRC cell proliferation by directly binding to the p53 promoter region and inhibiting p53 expression through H3K4me2 demethylation, which in turn inhibited the p53-p21-Rb pathway to regulate cell cycle progression (especially the G1/S phase transition) and inhibit apoptosis [25].

## 5. JMJD Family and CRC

### 5.1. KDM2 Family and CRC

Cao et al. reported for the first time that the expression level of KDM2A was elevated in human colorectal adenocarcinoma tissues by immunohistochemical assay, and further investigation revealed that KDM2A promoted colorectal adenocarcinoma cell proliferation and colony formation and was closely related to the expression of cyclin D1 [103]. Xi et al. demonstrated that the lncRNA LINC01278 enhanced KDM2A expression through competitive interaction with miR-143, thereby promoting CRC cell viability, migration, and invasion [104]. Liu et al. found that lncRNA TUG1 (positively regulated by transcription factor SP1) promoted the expression of KDM2A by inhibiting miR-421 and thus activating the ERK signaling pathway, thereby enhancing the proliferation and invasion of CRC cells and inhibiting apoptosis [105]. These effects could be reversed by knockdown of KDM2A [105]. KDM2A inhibits the transcription of rRNA genes by demethylating H3K36me2, thereby reducing ribosome synthesis [106]. A recent study demonstrated that depletion of KDM2A significantly enhanced the toxicity of 5-fluorouracil to CRC cells by upregulating rRNA transcription [107]. This potentiation was achieved by increasing the chances of 5-fluorouracil doping of rRNA, exacerbating the impairment of ribosome biosynthesis [107]. This finding provides a theoretical basis for the development of new combination therapeutic regimens, especially in those tumor types that are resistant to 5-fluorouracil, where KDM2A inhibitors may serve as an effective sensitizer.

Zacharopoulou et al. described that KDM2B regulated EZH2 and BMI1 in HCT116 cells [108]. KDM2B knockdown induced upregulation of protein levels of epithelial markers E-cadherin and zonula occludens-1 but downregulation of protein levels of mesenchymal markers N-cadherin and three small GTPases (RhoA, RhoB, and RhoC), suggesting that KDM2B may be involved in the EMT, cytoskeletal rearrangement, and migratory capacity of CRC cells [108].

### 5.2. KDM3 Family and CRC

The role of the KDM3 family in CRC has been increasingly revealed (Table 3). 15-Lipoxygenase-1 (15-LOX-1) is transcriptionally silenced in CRC and re-expression of 15-LOX-1 inhibited cancer cell proliferation and restored terminal cell differentiation and apoptosis [109]. Accumulating evidence suggests that 15-LOX-1 inhibits inflammation-driven colorectal tumorigenesis and suppresses signaling pathways that play a major role in chronic inflammation-promoting CRC, such as TNF-α, IL-1β/NF-κB, and IL-6/STAT3 pathways [110]. Zuo et al. showed that KDM3A is required for transcriptional activation of 15-LOX-1 in colon cancer via H3K9me2 demethylation in the 15-LOX-1 promoter [111]. Knockdown of KDM3A in Caco-2 cells reduced KDM3A recruitment to the 15-LOX-1 promoter with a marked increase in H3K9me2 levels [111]. However, the specific molecular mechanism through which KDM3A regulates 15-LOX-1 levels remains unrevealed. Uemura and colleagues established that KDM3A was an independent prognostic factor for CRC, and that KDM3A inhibition was associated with decreased proliferative activity and reduced invasion in CRC cell lines and tumor xenografts [112]. Padi et al. demonstrated that miR-627 increased histone H3K9me and inhibited proliferative factor expression by downregulating its target KDM3A in HT-29 cells [113]. Vitamin D induced miR-627 expression to downregulate KDM3A and consequently inhibited xenograft tumor growth in nude mice, suggesting that targeting KDM3A may have the same anti-tumor activities as vitamin D [113]. Li et al. reported that depletion of the KDM3 family (KDM3A, KDM3B, and KDM3C) inhibited self-renewal and chemoresistance in human colorectal CSCs [114]. Interestingly, KDM3 not only directly erased the repressive H3K9me2 marker but also contributed to the recruitment of mixed lineage leukemia 1 (an HMT) to promote H3K4 methylation and thus Wnt target gene transcription [114]. These results suggest that KDM3 is a key epigenetic regulator in the Wnt signaling pathway, providing a new potential target for CRC therapy. Consistently, Peng et al. reported that KDM3A demethylated H3K9me2 at c-Myc and matrix metalloproteinase 9 promoters and enhanced Wnt/β-catenin signaling by promoting β-catenin expression and interacting with β-catenin to enhance its transactivation to promote CRC progression [115].

Li et al. showed that KDM3A upregulated the chromatin remodeler ATRX (α-thalassemia/mental retardation syndrome X-linked) through H3K9me2 demethylation at the ATRX gene promoter [116]. Thus, the KDM3A-ATRX axis may represent a mechanism by which KDM3A promotes CRC. Liu et al. found that KDM3A expression was significantly increased in CRC metastatic lesions and correlated with poor histologic differentiation, advanced clinical stage, and short overall survival [117]. Mechanistic studies revealed that KDM3A functioned as an oncogene to regulate CRC cell migration and invasion through the regulation of EMT and matrix metalloproteinase [117]. Wang et al. demonstrated that KDM3A depletion led to the upregulation of H3K9me2 at TEA domain transcription factor 1 (TEAD1)-binding enhancers, which further resulted in decreased H3K27ac (acetylation), reduced TEAD1 binding to the enhancer, and impaired growth and migration of CRC cells [118]. In addition, KDM3A upregulated YAP1 (Yes-associated protein 1) expression and was associated with p300, suggesting that KDM3A expression is associated with YAP1 and Hippo target genes in CRC [118]. Jin et al. revealed that miR-22-3p targets KDM3A to regulate YAP1 expression to exert anti-tumor effects, suggesting that KDM3A regulates the Hippo signaling pathway to promote CRC progression [119]. Another study further demonstrated that under nutrient starvation, p300-dependent PHF5A acetylation was upregulated and induced alternative splicing to stabilize KDM3A mRNA and promote its protein expression, which promoted stress tolerance colon carcinogenesis in CRC cells [120]. Sui et al. suggested that two transcriptional regulators, MyoD family inhibitor and MDFI domain-containing, interact with KDM3A and together influence the transcription of several genes (e.g., the HIC1 oncogene) to promote the growth of CRC cells in vitro [121]. Oh et al. indicated that the oncogenic transcription factor ETV1 (ETS variant 1) bound directly to KDM3A, and both bound to the FOXQ1 (Forkhead Box Q1) promoter to co-regulate the ETV1 target gene FOXQ1 [122]. Overexpression of FOXQ1 partially reversed the inhibitory effect of ETV1 inhibition on HCT116 cells, suggesting that the ETV1/KDM3A-FOXQ1 axis may drive CRC tumorigenesis [122]. The authors subsequently showed that ETV1 co-controls BHLHE40 (basic helix–loop–helix family member e40) transcription with KDM3A and KDM4A, and that BHLHE40 downregulation inhibited the growth and clone formation activity of HCT116 cells, revealing that the ETV1/KDM3A/KDM4A-BHLHE40 axis may stimulate colorectal tumorigenesis [123].

Liu et al. showed that phosphatase of regenerating liver 3, a key metastatic gene in CRC, affected the activity of KDM3B and KDM4B as a key regulator of histone demethylation, and that the low expression of KDM3B was positively correlated with the lymph node status, Dukes classification, and TNM stage of CRC patients [124]. Chen et al. found that KDM3C was overexpressed in CRC tissues and positively correlated with metastasis and poor prognosis, and silencing KDM3C strongly inhibited CRC migration and invasion in vitro and in vivo [125]. Mechanistically, knockdown of KDM3C reduced the level of ATF2 (activating transcription factor 2) by regulating H3K9me2, thereby eliminating the oncogenicity caused by ATF2 overexpression [125].

### 5.3. KDM4 Family and CRC

The role of the KDM4 family in CRC has been extensively investigated (Table 4). Kim et al. showed that KDM4A interacts with the oncogene p53 in CRC cells in vitro and is co-recruited into the promoter of p21 [126]. Knockdown of KDM4A resulted in increased expression of p21 and pro-apoptotic Puma proteins, decreased levels of anti-apoptotic Bcl-2 protein, and reduced proliferation of CRC cells [126]. Song et al. reported that CXC chemokine receptors 4 and 7 could form heterodimers in vivo and promote colorectal tumorigenesis through KDM4A-mediated histone demethylation, leading to transcription of inflammatory factors and oncogenes [127].

Fu et al. showed that KDM4B was overexpressed in CRC cells in a hypoxia-inducible factor 1α-dependent manner under hypoxia and regulated CRC biological behaviors by upregulating the expression of a subset of hypoxia-inducible genes through demethylation of H3K9me3 [128]. The same authors subsequently showed that KDM4B knockdown induces DNA damage through ATM and Rad3-related pathways and STAT3 pathway activation, leading to CRC cell cycle arrest, apoptosis, and senescence [129]. Deng et al. further showed that the transcription factor CREB regulates the role of the KDM4B-STAT3 signaling in the DNA damage response by directly binding to a conserved region in the KDM4B promoter [130]. Fu et al. showed that glucose deprivation upregulated KDM4B expression (via ERK phosphorylation), increased the interaction between KDM4B and p-ERK, and also resulted in p-ERK phosphorylating KDM4B (which contributes to the stability of KDM4B) [131]. Knockdown of KDM4B significantly impaired colon cancer cell viability by increasing H3K9me3 levels on the glucose transporter 1 promoter [131]. Li et al. found that KDM4B activated the AKT signaling pathway by promoting TRAF6-mediated AKT ubiquitination to regulate glucose transporter 1 expression, which in turn promoted glucose metabolism and led to CRC progression [132]. Tan et al. further showed that depletion of KDM4B under glucose deficiency inhibited autophagy in CRC cells through H3K9me3 demethylation of the light chain 3 beta promoter, which reduced intracellular levels of specific amino acids (asparagine, phenylalanine, and histidine) and thus inhibited CRC cell survival [133]. Berry et al. showed that KDM4B forms a complex with β-catenin in vitro and in vivo, and that knockdown of KDM4B resulted in reduced expression of β-catenin/TCF4 target genes, suggesting that KDM4B contributes to colorectal tumorigenesis by supporting β-catenin-mediated gene transcription [134]. Sun et al. indicated that in vitro KDM4B silencing induced apoptosis in CRC cells by decreasing the expression of the anti-apoptotic gene Bcl-2 family and increasing the amount of cleaved caspase-8 involved in the death receptor-related apoptotic pathway [26]. Li et al. consistently showed that KDM4B inhibition significantly promoted mitochondrial apoptosis in CRC cells, and that these effects were induced partly by transcriptional repression of HCLS1 associated protein X-1 (an anti-apoptotic gene) through demethylation of H3K9me3 in the HCLS1 associated protein X-1 promoter region [135]. Liu et al. showed that enterotoxigenic *Bacteroides fragilis* increased stemness of CRC cells both in vivo and in vitro by upregulating KDM4B levels via the Toll-like receptor 4-nuclear factor of activated T cells 5-dependent pathway, which was mediated by upregulation of Nanog homeobox expression by KDM4B (via H3K9me3 demethylation on the promoter) [136]. Chen et al. reported that KDM4B interacted with ETS-related gene 1 and was recruited to the TC10-like promoter for H3K9 demethylation to activate transcription, whereas KDM4B knockdown attenuated migration and invasion of CRC cells [137].

Yamamoto et al. found that β-catenin bound to the KDM4C promoter and co-recruited to Jagged 1 promoter to maintain CSC-associated sphere formation capacity in CRC [138]. Wu et al. demonstrated that translocation of KDM4C protein into the nucleus decreased the levels of H3K9me3 and H3K36me3 of the lncRNA MALAT1 promoter and enhanced β-catenin signaling in CRC cells [139]. Liao et al. showed that ARID3B (AT-rich interaction domain-containing protein 3B) recruited KDM4C for H3K9me3 demethylation to activate Notch target genes, intestinal stem cell genes, and programmed death ligand 1 (PD-L1) to promote stem cell-like properties and immune escape in CRC cells [140]. Li et al. showed that exosomal circular RNA circPABPC1 functioned as an oncogene that recruited KDM4C to the high mobility group AT-hook 2 promoter for H3K9me3 demethylation, which initiated transcriptional processes and facilitated the progression of CRC liver metastasis [141]. Pu et al. showed that another circular RNA, circ_0000345, promoted CRC lung metastasis by enhancing the activation of the KDM4C/β-catenin signaling pathway through miR-205-5p, whereas a natural flavonoid compound, kaempferol, inhibited KDM4C/β-catenin signaling by decreasing circ_0000345 expression and suppressing CRC metastasis [30]. A recent study proposed that another natural compound, Avenanthramide A, binds directly to KDM4C leading to its degradation via the ubiquitin/proteasome pathway, resulting in the occupancy of the MIR17HG promoter by H3K9me3. Avenanthramide A enhanced the therapeutic effect of 5-fluorouracil by impairing the KDM4C/MIR17HG/GSK-3β negative feedback loop [142].

Peng et al. demonstrated that KDM4D was significantly upregulated in human CRC tissues and that KDM4D knockdown reduced proliferation, migration and invasion of mouse CRC cells [143]. Mechanistically, KDM4D physically interacts with β-catenin and activates transcription through H3K9me3 demethylation at the promoters of β-catenin target genes [143]. The authors further found that KDM4D enhanced glycolysis to promote CRC progression by activating the hypoxia-inducible factor 1 signaling pathway through interaction with the transcription factors SRY-Box Transcription Factor 9 and c-Fos, as well as hypoxia-inducible factor 1α [144]. Zhuo et al. demonstrated that KDM4D interacts with Gli2 to reduce the level of H3K9me3 at the promoter and thus promotes the expression of Hedgehog pathway target genes, and that KDM4D inhibitors work synergistically with Hedgehog pathway inhibitors to inhibit CRC cell proliferation and tumorigenesis [145]. Chen et al. found that KDM4D coactivated SP-1 to promote interferon gamma receptor 1 expression, which enhanced STAT3-Interferon regulatory factor 1 signaling and promoted PD-L1 expression through H3K9 demethylation for immune evasion in CRC [31]. In addition, the combination of KDM4D inhibitor and PD-L1 antibody could help to improve the anti-tumor efficacy of PD-L1 antibody [31].

### 5.4. KDM5 Family and CRC

The role of the KDM5 family in CRC is summarized in Table 5. Evensen et al. found that hypoxia-inducible factor 2α directly binds to hypoxia-responsive elements within the promoter region of cell migration-inducing protein, thereby inhibiting KDM5A function, leading to increased levels of H3K4me3 within the cell migration-inducing protein promoter and upregulation of cell migration-inducing protein, which drives CRC cell EMT and migration [146]. Shen et al. found that lncRNA NEAT1 inhibited KDM5A expression by binding to E2F transcription factor 1, and KDM5A downregulation activated the Wnt pathway through demethylation of H3K4me3 and activation of cullin 4A expression, which facilitated the progression of CRC [147]. Huang et al. found that the interaction between another lncRNA, TP73-AS1, and KDM5A resulted in reduced levels of H3K4me3 at the TP73 promoter in CRC cells, affecting TP73 transcription and partially explaining the malignant behavior in CRC [148].

Ohta et al. found that KDM5B depletion led to loss of epithelial differentiation and inhibited CRC cell growth, which was associated with induced cellular senescence [149]. Huang et al. revealed that KDM5B modulated the Wnt/β-catenin signaling pathway to promote CRC cell proliferation by regulating Caudal Type Homeobox 2 expression through H3K4me3 demethylation [150]. Yan et al. found that KDM5B reduced CC chemokine ligand 14 expression through demethylation of H3K4me3 to activate Wnt/β-catenin, and that KDM5B knockdown or overexpression of CC chemokine ligand 14 inhibited the proliferation and invasiveness of CRC cells [151].

Lin et al. found that KDM5C overexpression significantly reduced the half maximal inhibitory concentrations (IC50) of oxaliplatin and irinotecan and promoted multidrug resistance in CRC by reducing ATP Binding Cassette Subfamily C Member 1 expression through H3K4me3 demethylation at the transcriptional start site of ATP Binding Cassette Subfamily C Member 1 [152]. Chen et al. found that RNA N6-adenosylmethyltransferase methyltransferase-like 14 inhibited CRC progression through SRY-Box Transcription Factor 4-mediated EMT and PI3K/Akt signaling, whereas KDM5C inhibited methyltransferase-like 14 transcription through H3K4me3 demethylation at the methyltransferase-like 14 promoter [153]. Yu et al. showed that KDM5C inhibited prefoldin subunit 5 transcription through H3K4me3 demethylation at its promoter, which enhanced the transcriptional activity of c-Myc [154]. Knockdown of KDM5C inhibited the oncogenicity of CRC cells and increased autophagy and apoptosis [154].

Liu et al. found that KDM5D overexpression in males significantly inhibited CRC growth and metastasis in vitro and in vivo [155]. Mechanistically, KDM5D inhibited E2F transcription factor 1 expression through H3K4me3 demethylation, thereby inhibiting FKBP Prolyl Isomerase 4 transcription and exerting anti-tumor and anti-metastasis effects in CRC [155]. A recent influential study found that mutation KRAS-STAT4-mediated upregulation of Y-chromosome KDM5D largely explains the sex difference in KRAS-mutated CRC (higher metastasis and mortality rates in male CRC) by disrupting the adhesion properties of cancer cells and tumor immunity, providing a viable therapeutic strategy to reduce the metastatic risk of CRC in men with KRAS mutations [28]. Chen et al. reported that the zinc finger CCHC domain-containing protein 4 (a m6A methyltransferase)–lncRNA GHRLOS-KDM5D axis regulated CRC progression in vitro and in vivo, and that KDM5D overexpression impaired CRC cell proliferation, migration, and invasion [156].

### 5.5. KDM6 Family and CRC

The role of the KDM6 family in CRC is summarized in Table 6. Zha et al. identified that KDM6A initiated E-cadherin expression in HCT-116 cells by demethylating H3K27me3, and overexpression of KDM6A inhibited migration and invasion of HCT-116 cells [157]. In addition, KDM6A synergistically activated E-cadherin transcription by increasing H3K27ac in the E-cadherin promoter [157]. Zhou et al. found that ten-eleven translocation 1 knockdown induced EMT in DLD1 cells and increased cancer cell growth, migration, and invasion by increasing EZH2 expression and decreasing KDM6A expression, leading to repression of the target gene E-cadherin [158]. However, Tang et al. found that KDM6A was upregulated in CRC tissues, and knockdown of KDM6A significantly inhibited CRC cell proliferation and led to cell cycle arrest by decreasing the expression of KIF14 and pAKT and increasing the expression of p21 [159]. Wang et al. found that oxaliplatin significantly induced the expression of KDM6A and KDM6B, whereas KDM6A/6B depletion or GSK-J4 (a KDM6A/6B inhibitor) treatment elevated the level of H3K27me3 at the transcription start site of NOTCH2, which significantly enhanced the drug sensitivity of oxaliplatin [27]. Zhang et al. similarly demonstrated that the inhibition of KDM6A/B using GSK-J4 induced global enhancer reprogramming, attenuated the malignant phenotype of CRC cells in vitro and in vivo, sensitized them to chemotherapy, and suppressed tumor-initiating cell characterization and stemness-associated gene signatures [29]. Ji et al. demonstrated that PCGF1 binds to the promoters of CRC stem cell markers and maintains the CRC stem cell-like phenotype by activating their transcription in part through the reduction in KDM6A-mediated H3K27me3 [160]. Du et al. found that KDM6A deficiency promotes the survival and accumulation of the immunosuppressive myeloid-derived suppressor cells through the secretion of tyrosine via methylation by phenylalanine hydroxylase, thereby facilitating immune escape [161]. Luo et al. identified that the CUL4B-DDB1-COP1 complex acts as a functional E3 ligase targeting KDM6A for degradation to promote CRC progression, and KDM6A deletion in intestinal epithelial cells enhances susceptibility to tumorigenesis in a spontaneous mouse CRC model [162]. Sharma et al. found that p53-deficient CRC cells exhibited higher migratory and invasive capacities during TGF-β-induced EMT by inducing synergistic interactions between KDM6A and KDM6B and EMT transcription factors such as SNAI1 and SNAI2 [163]. A recent study found that KDM6A knockdown in CRC cells enhanced glycolysis by promoting the binding of hypoxia-inducible factor 1α to the lactate dehydrogenase A promoter, thereby promoting lactate dehydrogenase A expression and lactate production [164].

Low KDM6B is a predictor of poor prognosis in CRC patients [165]. Pereira et al. found that the vitamin D metabolite 1,25(OH)2D3 induced KDM6B expression in colon cancer cells in a vitamin D receptor-dependent manner, whereas knockdown of KDM6B attenuated the inhibitory effects of 1,25(OH)2D3 on the Wnt/β-catenin signaling pathway and led to increased expression of the EMT markers [166]. Nagarsheth et al. found that KDM6B-mediated demethylation of H3K27me3 with the PRC2 component silenced Th1-type chemokines and suppressed effector T cell-mediated immunity, controlling immunosuppression in colon cancer [167]. Lian et al. found that the Notch signaling pathway regulator Notch intracellular domain formed a complex with KDM6B and bound to the EPHB4 enhancer region, which reduced H3K27me3 levels and promoted proliferation and invasion by promoting EPHB4 expression [168]. In a recent study, Guo et al. showed that sphingosine-1-phosphate receptor 2 upregulated nuclear TWIST1 (twist family bHLH transcription factor 1) through activation of the Hippo/TEAD1-TWIST1 pathway, and that nuclear TWIST1 interacted with the KDM6B-RNA Pol II complex, leading to dihydropyrimidine dehydrogenase transcription through H3K27me3 demethylation, which mediated 5-fluorouracil resistance in CRC [169].

**Table 6 cimb-47-00267-t006:** Overview of the roles and mechanisms of the KDM6 family in CRC.

References	HDM	Histone Demethylation Sites	Gene Expression Profile	Mechanisms of Action	Impact on CRC
[157]	KDM6A	H3K27me3	NA	Regulation of E-cadherin expression	Inhibition of CRC cell migration and invasion
[158]	KDM6A	H3K27me3	NA	Regulation of E-cadherin expression	Inhibition of EMT and tumor invasion
[159]	KDM6A	NA	Upregulated	Regulation of KIF14 and pAKT expression	Promotion of CRC cell proliferation
[27]	KDM6A/B	H3K27me3	NA	Interruption of oxaliplatin-induced NOTCH signaling	KDM6A/6B depletion enhanced oxaliplatin-induced apoptosis
[29]	KDM6A/B	H3K27me3	NA	Inducing global enhancer reprogramming	KDM6 inhibition attenuated the malignant phenotype of CRC cells, sensitized them to chemotherapy, and suppressed tumor-initiating cells’ properties and stemness-associated gene signatures
[160]	KDM6A	H3K27me3	NA	Mediating CRC stem cell marker gene activation	Mediating CRC stem cell proliferation
[161]	KDM6A	NA	NA	Promotion of survival and accumulation of myeloid-derived suppressor cells	KDM6A deficiency promoted immune escape and CRC progression
[162]	KDM6A	H3K27me3	Downregulated	Targeting EMP1 and AUTS2	KDM6A downregulation promoted CRC progression
[163]	KDM6A/B	H3K27me3	NA	Mediating EMT	NA
[164]	KDM6A	NA	NA	Enhancement of glycolysis by promoting the binding of hypoxia-inducible factor 1α to the lactate dehydrogenase A promoter	KDM6A deletion promoted CRC progression
[166]	KDM6B	NA	NA	Regulation of the EMT and Wnt/b-catenin pathways	Mediating the inhibitory effect of 1,25(OH)2D3 on CRC
[167]	KDM6B	H3K27me3	NA	Increased expression of Th1-type chemokines	Mediating immune evasion in CRC
[168]	KDM6B	H3K27me3	NA	Promoting EPHB4 expression	Promoting proliferation and invasion
[169]	KDM6B	H3K27me3	NA	Upregulation of dihydropyrimidine dehydrogenase expression	Mediating 5-fluorouracil resistance in CRC

Abbreviations: CRC, colorectal cancer; HDM, histone demethylase; KIF14, kinesin family member 14; NA, not available; EMT, epithelial–mesenchymal transition.

### 5.6. KDM7 Family and CRC

The KDM7 family remains poorly studied in CRC. lv et al. showed that KDM7B was upregulated in CRC tissues and that loss of KDM7B significantly inhibited CRC cell proliferation and migration and promoted CRC cell apoptosis [170]. Sun et al. recently showed that the RNA-binding protein hnRNPA2B1 promoted CRC progression by increasing the m6A level of circCDYL, thereby decreasing the binding of circCDYL to EIF4A3 and enhancing the expression of KDM7B [171]. In a recently published study, Liu et al. identified KDM7B as an oncogenic factor in KRAS- or BRAF-mutant CRC by genome-wide CRISPR screening, and targeting KDM7B significantly improved the efficacy of anti-PD1 therapy [172]. Mechanistically, KDM7B upregulated the expression of PD-L1, KRAS, BRAF, and c-Myc and promoted immune escape and tumor progression by increasing the levels of H3K4me3 and H3K27ac and decreasing the level of H3K9me2 [172]. Lee et al. showed that KDM7C, which was downregulated in CRC tissues, is an epigenetic regulator of p53 and positively correlates with p21 expression, which transforms transcriptionally favorable chromatin through H3K9me2 demethylation and ensures p53-mediated cell death in response to chemotherapy [173].

### 5.7. KDM8 Family and CRC

Only one study preliminarily explored the role of KDM8 in CRC. Zhang et al. found that KDM8 was upregulated in human colon cancer tissues, and in vitro KDM8 depletion significantly inhibited colon cancer cell proliferation, migration and invasion, suggesting that KDM8 may be a potential oncogene [174].

### 5.8. RIOX1/2 and CRC

The role of RIOX1/2 in CRC remains understudied. Nishizawa et al. preliminarily showed that RIOX1 was selectively expressed in human CRC tissues compared to normal tissues and was associated with metastatic malignant behavior and prognosis, whereas in vitro RIOX1 knockdown inhibited CRC cell proliferation, migration, and anti-apoptotic activity [175]. Teye et al. demonstrated that RIOX2 expression was increased in human CRC tissues and in vitro CRC cells, and mina53 knockdown inhibited CRC cell proliferation in vitro [176]. Fujino et al. showed that positive nuclear localization of RIOX2 was associated with recurrence and poor prognosis in CRC patients after adjuvant chemotherapy [177].

## 6. Therapeutic Opportunities for Targeting HDM in CRC

### 6.1. LSD Family Inhibitor and CRC

Some KDM1A inhibitors have demonstrated therapeutic potential in preclinical CRC models (Table 7), but their clinical translational potential in CRC patients remains poorly explored. Huang et al. showed that several polyamine/oligoamine analogs were potent KDM1A inhibitors and induced the re-expression of several aberrantly silenced genes in CRC cells [178,179]. Sorna et al. identified a new N’-(1-phenylethyl)-benzohydrazide series by high-throughput virtual screening, compound **12**, which was a potent, specific, and reversible inhibitor of KDM1A and suppressed CRC cell proliferation and survival [180]. Another compound identified by high-throughput docking, L05, was shown to be a KDM1A inhibitor and inhibited CRC cell migration without significant cytotoxicity [181]. Li et al. showed that polyethylene glycolated (PEGylated) 17i nanoassemblies were superior to the KDM1A inhibitor compound **17i** in terms of therapeutic efficiency, anti-tumor immune response, and systemic toxicity in xenografted CRC mice [182]. Several small-molecule KDM1A inhibitors, such as CBB1003 [88], ORY-1001 [89], GSK2879552 [89], ZY0511 [96,97,98], and tranylcypromine/tranylcypromine hemisulfate [95,99], were shown to inhibit CRC cell growth or migration by modulating specific gene or pathway expression in in vitro or in vivo CRC models.

### 6.2. JMJD Family Inhibitor and CRC

Chen et al. demonstrated that treatment of CRC cells with daminozide, a KDM2A small-molecule inhibitor, enhanced de novo RNA transcription and increased 5-fluorouridine toxicity and apoptotic cell death [107].

A KDM3A inhibitor, a family of carboxamide-substituted benzhydryl amines (CBA-1), inhibited Wnt target genes and CRC cell proliferation in vitro [183]. IOX1 (8-hydroxyquinoline-5-carboxylic acid) has been demonstrated to significantly suppress Wnt target gene transcription and CRC tumorigenesis by inhibiting KDM3 activity [184]. Zaman et al. recently designed and synthesized several PROTACs (Proteolysis Targeting Chimeras) using the broad-spectrum KDM3 small-molecule inhibitor IOX1 as a warhead, of which compound **4** showed a remarkable ability to inhibit Wnt signaling for the elimination of colorectal CSCs and inhibition of tumor growth, with a potency that was approximately 10- to 35-fold higher than that of IOX1 [185].

Franci et al. showed that PKF118-310, as a TCF4/β-catenin pathway antagonist, is a novel KDM4A inhibitor, and that its treatment inhibited the proliferation of HCT-116 cells and affected the cell cycle and induced apoptosis in vitro [186]. Kim et al. showed that benzo[b]tellurophenes (compound **1c**) were KDM4-specific inhibitors and induced apoptosis in CRC cells in vitro [187]. Chen et al. showed that compound **6** (QC6352) was a potent KDM4 family inhibitor that inhibited colony formation and suppressed cell viability in a patient-derived colon cancer organoid model [188]. Liao et al. demonstrated that treatment with the KDM4C inhibitor SD70 and the KDM4A/B inhibitor NSC636819 attenuated the expression of target genes induced by ARID3B (which promotes CRC stem cell-like features) [140]. Recent studies have shown that two natural compounds, kaempferol and Avenanthramide A, directly inhibit KDM4C expression and act as KDM4C inhibitors against CRC tumorigenesis [30,142]. Zhuo et al. showed that the KDM4D inhibitor 5-c-8HQ or aspirin synergized with the Hedgehog inhibitor vismodegib to inhibit CRC cell proliferation and tumorigenesis [145]. 5-c-8HQ treatment also helped to improve the anti-CRC tumor efficacy of PD-L1 antibody in mice [31]. Fang et al. reported that a new class of highly selective inhibitors of KDM4D, 24s, significantly inhibited the proliferation and migration of CRC cells in vitro [189]. A recent study described a novel small-molecule KDM4 pan-inhibitor, TACH101, that selectively targets KDM4A-D and has no effect on the other KDM families [190]. TACH101 showed potent antiproliferative activity against a variety of malignancies including CRC in organoid models [190].

KDM5A may exhibit dual roles as a tumor-promoting or tumor-suppressing agent in CRC in different microenvironments. Huang et al. showed that lncRNA TP73-AS1 interacts with KDM5A to promote CRC malignant behavior by affecting TP53 transcription, whereas treatment with the KDM5A inhibitor HY-100014 reduced KDM5A occupancy at the TP53 promoter [148]. Notably, Shen et al. showed that treatment of HCT-116 cells with the KDM5 pan-inhibitor CPI-455 increased H3K4me3 levels which in turn led to increased cullin 4A expression and may promote CRC progression in a Wnt-dependent manner [147]. Chen et al. showed that in vitro treatment of CRC cells with the KDM5C inhibitor KDM5A-IN-1 increased the H3K4me3 level of the tumor suppressor methyltransferase-like 14 promoter, thereby promoting its expression [153].

Treatment with the small-molecule KDM6A/6B inhibitor GSK-J4 increased chemotherapy sensitivity and attenuated the malignant phenotype in CRC cells by increasing H3K27me3 levels [27,29]. However, Nagarsheth et al. showed that GSK-J4 treatment resulted in a reduction in the Th1-type chemokines C-X-C Motif Ligand 9 and 10 in colon cancer cells, decreasing anti-tumor immunity [167]. Liu et al. recently found that treatment with the KDM7B inhibitor daminozide reduced the proliferation of CRC cells and lowered the expression levels of the oncogenes KRAS, BRAF, and c-Myc in vitro [172].

Overall, there are several JMJD family inhibitors that have demonstrated therapeutic potential in experimental CRC models (Table 8), but clinical trials are lacking. HDM inhibitors (including the LSD family and the JMJD family) and their corresponding HDM targets in CRC are presented in Figure 3. However, it is encouraging that clinical trials are already underway exploring the efficacy and safety of HDM inhibitors (mainly KDM1A) in other cancer types. A recent review provides a detailed summary of clinical trials of KDM1A inhibitors in other cancers, including small-cell lung cancer, hematologic malignancies, acute myeloid leukemia, myelodysplastic syndromes, Ewing’s sarcoma, solid tumors, myelofibrosis, essential thrombocythemia, prostate cancer, sarcomas, and ovarian-related tumors (please refer to this review for specific details and trial numbers) [191]. All clinical trials are in Phase I, I/II, or II [191]. However, there is still a lack of clinical trials conducted on the effects of JMJD family inhibitors in cancer.

## 7. Conclusions and Future Directions

In recent years, the regulatory role of HDMs in CRC has been gradually revealed. It has been found that HDMs are involved in CRC occurrence, progression and metastasis by affecting the tumor microenvironment and key signaling pathways through epigenetic modifications. These HDMs interact with diverse factors (e.g., transcription factors) which in turn affect the expression of key genes by regulating target gene transcription through histone demethylation at different sites and may be regulated by certain upstream genes. In addition, HDMs may have dual roles in CRC—either promoting a malignant tumor phenotype or inhibiting progression—depending on their target genes and the molecular networks they regulate. Experimental evidence suggests that some small-molecule inhibitors targeting HDMs may be an emerging strategy for CRC treatment; however, clinical translation and confirmation in clinical trials are lacking. Some of the methodological limitations of the existing studies are worth noting. Many preclinical studies do not include clinical patient cohorts or use small patient xenograft cohorts or limited cell line models that may not adequately reflect the molecular diversity of CRC. Clinical relevance often relies on retrospective The Cancer Genome Atlas data with uneven racial/ethnic representation. Many HDM inhibitor-related studies have used monolayer cultures, neglecting the effect of the three-dimensional tumor microenvironment on drug response. There may be several directions that need to be further explored in the future. First, the development of targeted inhibitors for HDMs is limited by substrate selectivity and off-target effects. Combining structural biology and artificial intelligence is needed to design highly selective inhibitors. The role of HDM-specific inhibitors in CRC remains understudied, and further studies are needed to elucidate their role and translational potential in CRC to advance clinical trials. Single-cell sequencing and spatial transcriptomic technologies can reveal the dynamic roles of HDMs in different regions and cellular subpopulations of CRC for the resolution of microenvironmental and spatiotemporal heterogeneity. The association of HDM expression profiles with prognosis and chemotherapy resistance in CRC needs to be established to facilitate biomarker development and clinical translation. The role of certain HDMs in CRC is unclear, and further mechanistic studies are needed to elucidate and explore more possible roles of HDMs in CRC, such as modulation of gut microbiota and energy metabolism. In conclusion, the complex role of HDMs in CRC provides a new target for precision therapy, but its clinical translation and mechanism studies still need to be further explored. Future studies should integrate multi-omics, preclinical modeling and interdisciplinary technologies to promote HDMs from mechanism exploration to clinical application.

## Figures and Tables

**Figure 1 cimb-47-00267-f001:**
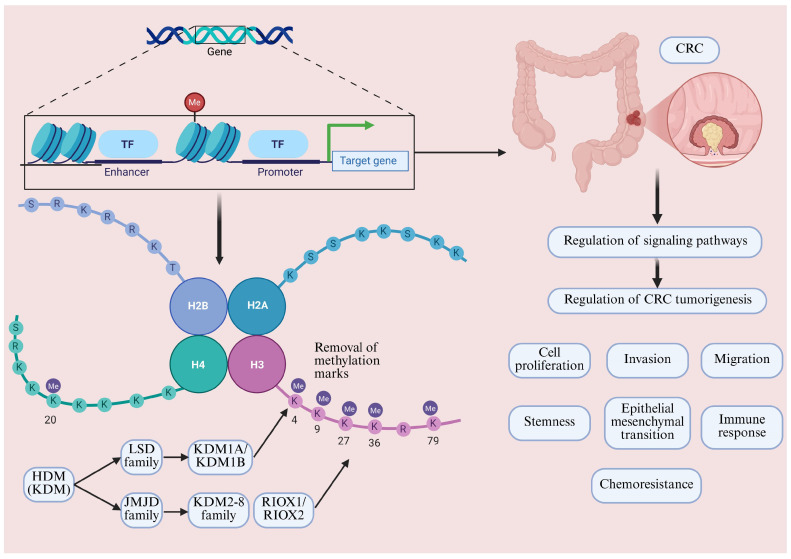
Mechanistic overview of HDMs in CRC.

**Figure 2 cimb-47-00267-f002:**
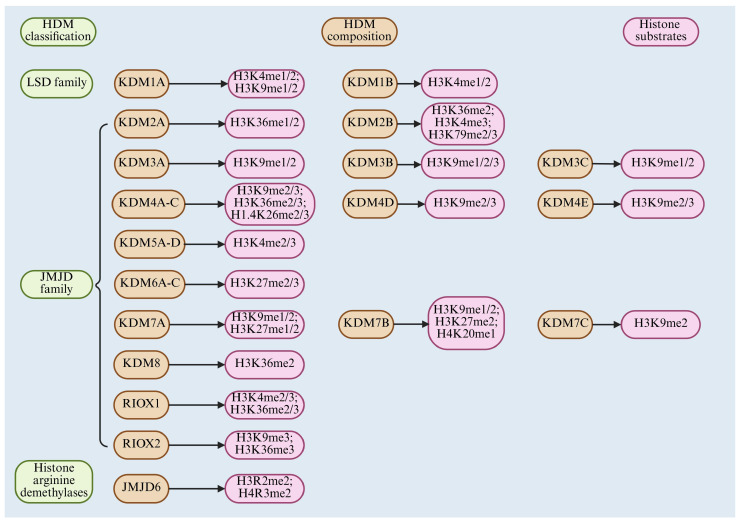
HDM classification, composition, and substrate overview.

**Figure 3 cimb-47-00267-f003:**
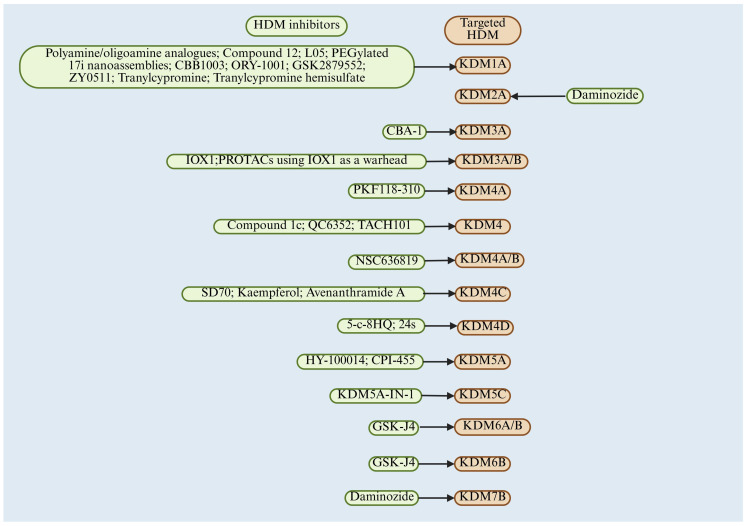
HDM inhibitors and their corresponding HDM targets in CRC.

**Table 1 cimb-47-00267-t001:** HDMs with demethylation activity and their aliases and histone substrates.

HDM	Aliases	Histone Substrates
Confirmed activity		
KDM1A	LSD1, AOF2, BHC110	H3K4me1/2; H3K9me1/2
KDM1B	LSD2, AOF1	H3K4me1/2
KDM2A	JHDM1A, FBXL11	H3K36me1/2
KDM2B	JHDM1B, FBXL10	H3K36me2; H3K4me3; H3K79me2/3
KDM3A	JMJD1A, JHDM2A, TSGA	H3K9me1/2
KDM3B	JMJD1B	H3K9me1/2/3
KDM4A	JMJD2A, JHDM3A	H3K9me2/3; H3K36me2/3; H1.4K26me2/3
KDM4B	JMJD2B, JHDM3B	H3K9me2/3; H3K36me2/3; H1.4K26me2/3
KDM4C	JMJD2C, JHDM3C, GASC1	H3K9me2/3; H3K36me2/3; H1.4K26me2/3
KDM4D	JMJD2D, JHDM3D	H3K9me2/3
KDM5A	JARID1A, RBP2	H3K4me2/3
KDM5B	JARID1B, PLU1	H3K4me2/3
KDM5C	JARID1C, SMCX	H3K4me2/3
KDM5D	JARID1D, SMCY	H3K4me2/3
KDM6A	UTX	H3K27me2/3
KDM6B	JMJD3	H3K27me2/3
KDM7A	KIAA1718, JHDM1D	H3K9me1/2; H3K27me1/2
KDM7B	PHF8, JHDM1F	H3K9me1/2; H3K27me2; H4K20me1
RIOX1	NO66, MAPJD, C14orf169	H3K4me2/3; H3K36me2/3
Controversial activity		
KDM3C	JMJD1C	H3K9me1/2
KDM4E	JMJD2E	H3K9me2/3
KDM6C	UTY	H3K27me2/3
KDM7C	PHF2, JHDM1E	H3K9me2
KDM8	JMJD5	H3K36me2
RIOX2	MINA, Mdig, NO52; MINA53	H3K9me3; H3K36me3
JMJD6	PTDSR, KIAA0585	H3R2me2; H4R3me2

**Table 2 cimb-47-00267-t002:** Overview of the roles and mechanisms of the LSD family in CRC.

References	HDM	Histone Demethylation Sites	Gene Expression Profile	Mechanisms of Action	Impact on CRC
[83]	KDM1A	NA	NA	Increased expression of target genes	KDM1A deletion led to reduced cell proliferation in vitro and in vivo
[84]	KDM1A	NA	Upregulated	Negative correlation with E-cadherin expression	Positively correlated with lymph node and distant metastases among CRC patients
[85]	KDM1A	H3K4me2	Upregulated	Downregulation of E-cadherin expression	Inhibition of KDM1A impaired proliferation and invasiveness and induced apoptosis in colon cancer cells
[86]	KDM1A	NA	Upregulated	Activation of the Wnt/β-catenin signaling pathway through downregulation of the pathway antagonist DKK1	KDM1A knockout led to lower tumorigenicity of CRC cells in vivo and in vitro
[87]	KDM1A	NA	NA	NA	KDM1A knockdown impaired stemness of CD133+ CRC cells
[88]	KDM1A	NA	Upregulated	KDM1A inhibitor treatment downregulated LGR5 levels and inactivated the Wnt/β-catenin pathway	KDM1A inhibitor treatment inhibited CRC cell proliferation and colony formation
[89]	KDM1A	NA	Upregulated	CSC stemness maintenance, cytoskeletal remodeling and altered mitochondrial function	Inhibition of KDM1A impaired migration and invasion of CRC cells
[90]	KDM1A	H3K4me2	NA	Inhibition of P21 and Krüppel-like factor 2 expression by binding to the lncRNA HOXA-AS2	Promotion of CRC cell proliferation
[91]	KDM1A	H3K4me2	NA	Interaction with lncRNA FEZF1-AS1 and inhibition of Krüppel-like factor 2 expression	Influence on CRC cell proliferation
[92]	KDM1A	H3K4me2	NA	Silencing of p21 and PTEN expression by binding to lncRNA DUXAP1	Promotion of CRC cell growth and reduction in apoptosis
[93]	KDM1A	H3K4me2	NA	Silencing of ASXL1 expression by LINC00586 recruitment to the ASXL1 promoter	Inhibition of CRC cell viability, migration, EMT, and in vivo tumorigenesis
[94]	KDM1A	NA	NA	Increased stability of RIOK1	Influencing tumor growth and metastasis
[95]	KDM1A	H3K4me2	Upregulated	Suppression of EMT	Promotion of CRC cell invasiveness
[96]	KDM1A	H3K4me1/2	NA	KDM1A inhibition increased expression of the mTORC1 suppressor DNA Damage Inducible Transcript 4	Inhibition of KDM1A suppressed CRC cell proliferation
[97]	KDM1A	NA	Upregulated	Inhibition of Wnt/β-catenin signaling and DNA synthesis pathways	KDM1A inhibition enhanced the anticancer effect of 5-fluorouracil
[98]	KDM1A	NA	NA	Altered lipid metabolism	NA
[99]	KDM1A	H3K9me2	Upregulated	Upregulation of tetraspanin 8 expression to promote EMT	Promotion of CRC cell proliferation and migration
[100]	KDM1A	NA	NA	KDM1A deletion blocked enteroendocrine cell progenitor cell formation	KDM1A deficiency reduced tumor growth and metastasis
[101]	KDM1A	NA	NA	Enhancement of the enteroendocrine cell specification regulator STAT3 chromatin binding	Influence on primary tumor growth and lung metastasis
[102]	KDM1A	NA	NA	Reprogramming tumor-associated macrophage subtypes	NA
[25]	KDM1B	H3K4me2	Upregulated	Inhibition of p53 activity	Promotion of CRC cell proliferation and inhibition of apoptosis

Abbreviations: LSD, lysine-specific demethylase; CRC, colorectal cancer; HDM, histone demethylase; NA, not available; EMT, epithelial–mesenchymal transition; LGR5, leucine-rich repeat-containing G-protein-coupled receptor 5.

**Table 3 cimb-47-00267-t003:** Overview of the roles and mechanisms of the KDM3 family in CRC.

References	HDM	Histone Demethylation Sites	Gene Expression Profile	Mechanisms of Action	Impact on CRC
[111]	KDM3A	H3K9me2	NA	Transcriptional activation of 15-LOX-1	NA
[112]	KDM3A	NA	Upregulated	NA	KDM3A deletion inhibited CRC cell proliferation and invasion
[113]	KDM3A	H3K9me2	NA	Mediating the anti-tumor activity of vitamin D as a direct target of miR-627	Downregulation of KDM3A suppressed proliferative factor expression and xenograft tumor growth
[114]	KDM3A/B	H3K9me2	NA	Promotion of Wnt target gene transcription	KDM3 depletion inhibited tumorigenic growth and chemoresistance in CSCs
[115]	KDM3A	H3K9me2	Upregulated	Enhancement of Wnt/β-catenin signaling	Promotion of CRC cell proliferation, migration, and invasion
[116]	KDM3A	H3K9me2	Upregulated	Upregulation of ATRX expression	Downregulation of KDM3A inhibited clonogenic activity in CRC cells
[117]	KDM3A	NA	Upregulated	Regulation of EMT and matrix metalloproteinase	Promotion of CRC cell migration and invasion
[118]	KDM3A	H3K9me2	NA	Positive regulators of hippo target genes	KDM3A depletion led to impaired growth and migration of CRC cells
[119]	KDM3A	NA	Upregulated	Regulation of the HIPPO signaling pathway as a target of miR-22-3p	Promotion of proliferation, invasion and migration of CRC cells
[120]	KDM3A	H3K9me2	NA	Promoted stress resistance in CRC cells	Promotion of colon carcinogenesis
[121]	KDM3A	NA	NA	Binding to MDFI domain-containing and MDFI domain-containing to influence transcription of several genes	Promotion of CRC cells growth in vitro
[122]	KDM3A	NA	NA	Binding to ETV1 to co-regulate FOXQ1	Driving colorectal tumorigenesis
[123]	KDM3A	NA	NA	Co-control of BHLHE40 transcription with ETV1 and KDM4A to upregulate KLF7 and ADAM19	NA
[124]	KDM3B	H3K9me3	Downregulated	NA	May function as a tumor suppressor
[125]	KDM3C	H3K9me2	Upregulated	Regulation of ATF2 expression	Silencing of KDM3C suppressed CRC migration and invasion in vitro and in vivo

Abbreviations: CRC, colorectal cancer; HDM, histone demethylase; 15-LOX-1, 15-lipoxygenase-1; NA, not available; ATRX, α-thalassemia/mental retardation syndrome X-linked; EMT, epithelial–mesenchymal transition; ETV1, ETS variant 1; ATF2, activating transcription factor 2; FOXQ1, Forkhead Box Q1; BHLHE40, basic helix–loop–helix family member e40.

**Table 4 cimb-47-00267-t004:** Overview of the roles and mechanisms of the KDM4 family in CRC.

References	HDM	Histone Demethylation Sites	Gene Expression Profile	Mechanisms of Action	Impact on CRC
[126]	KDM4A	H3K9me3	NA	Recruited with p53 to the promoter of p21	KDM4A depletion induced proliferation reduction and apoptosis in CRC cells
[127]	KDM4A	H3K9me3 and H3K36me3	Upregulated	Induction of transcription of inflammatory factors and oncogenes	Inhibition of KDM4A prevented CRC
[128]	KDM4B	H3K9me3	Upregulated	Upregulation of hypoxia-inducible genes	Promoting malignant phenotypes in CRC cells
[129]	KDM4B	H3K9me3	NA	Regulation of the DNA damage response	Mediation in cancer cell survival and tumor growth
[130]	KDM4B	NA	Upregulated	Promoting DNA damage response through STAT3 signaling	KDM4B silencing sensitized tumor cells to irradiation
[131]	KDM4B	H3K9me3	Upregulated	Upregulation of glucose transporter 1 to regulate glucose uptake	KDM4B knockdown impaired colon cancer cell viability
[132]	KDM4B	NA	Upregulated	Regulation of glucose transporter 1 expression through the AKT signaling pathway to promote glucose uptake and ATP production	Promotion of CRC proliferation
[133]	KDM4B	H3K9me3	NA	Inhibiting autophagy and reducing intracellular levels of specific amino acids in CRC cells by regulating light chain 3 beta	Downregulation of KDM4B inhibited CRC cell survival
[134]	KDM4B	NA	Upregulated	Promoting β-catenin-mediated gene transcription	Promoting CRC cell growth and clone formation
[26]	KDM4B	NA	NA	Regulation of mitochondria and death receptor-related pathways	KDM4B silencing induced apoptosis
[135]	KDM4B	H3K9me3	Upregulated	Activation of HCLS1-associated protein X-1 transcription to mediate mitochondrial apoptosis	Inhibition of KDM4B promoted CRC cell apoptosis
[136]	KDM4B	H3K9me3	NA	Upregulation of Nanog homeobox expression	Modulation of CSC properties
[137]	KDM4B	H3K9me2/3	Upregulated	Interact with ETS-related gene 1 to activate TC10-like transcription	KDM4B knockdown attenuated CRC cell migration and invasion
[138]	KDM4C	NA	Upregulated	Mediating β-catenin-dependent transcription of Jagged-1	KDM4C knockdown eliminated colonosphere formation from CRC cells
[139]	KDM4C	H3K9me3 and H3K36me3	Upregulated	Upregulation of MALAT1 and enhancement of β-catenin signaling pathway activity	Promoting CRC metastasis in vitro and in vivo
[140]	KDM4C	H3K9me3	NA	Activation of Notch target genes, intestinal stem cell genes and PD-L1	Promoting stem cell-like properties and immune escape in CRC cells
[141]	KDM4C	H3K9me3	Upregulated	Initiation of high mobility group AT-hook 2 transcription	Regulating EMT and cancer metastasis
[30]	KDM4C	NA	NA	Promoting β-catenin signaling	Regulation of CRC metastasis
[142]	KDM4C	H3K9me3	Upregulated	Regulation of MIR17HG transcription	Mediating the CRC chemotherapy response
[143]	KDM4D	H3K9me3	Upregulated	Interacting with β-catenin to activate transcription of its target genes	KDM4D knockdown reduced CRC cell proliferation, migration, invasion, and xenograft tumor formation
[144]	KDM4D	H3K9me3	NA	Enhancement of glycolysis by activating the hypoxia-inducible factor 1 signaling pathway	KDM4D knockdown inhibited CRC cell proliferation, migration, invasion, and xenograft growth and metastasis
[145]	KDM4D	H3K9me3	Upregulated	Interacting with Gli2 to promote expression of Hedgehog target genes	KDM4D knockdown inhibited CRC growth and metastasis
[31]	KDM4D	H3K9me3	Upregulated	Enhanced STAT3- Interferon regulatory factor 1 signaling and promoted PD-L1 expression	Promoting CRC immune escape

Abbreviations: CRC, colorectal cancer; HDM, histone demethylase; NA, not available; PD-L1, programmed death ligand 1; EMT, epithelial–mesenchymal transition.

**Table 5 cimb-47-00267-t005:** Overview of the roles and mechanisms of the KDM5 family in CRC.

References	HDM	Histone Demethylation Sites	Gene Expression Profile	Mechanisms of Action	Impact on CRC
[146]	KDM5A	H3K4me3	Upregulated	Upregulation of cell migration-inducing protein expression	Mediating CRC cell migration
[147]	KDM5A	H3K4me3	NA	Inhibition of cullin 4A expression	KDM5A downregulation mediated malignant behavior in CRC cells
[148]	KDM5A	H3K4me3	NA	Interact with TP73-AS1 to affect TP73 transcription	Mediating malignant behavior in CRC
[149]	KDM5B	H3K4me3	Upregulated	Induction of cellular senescence	KDM5B depletion led to loss of epithelial differentiation and inhibition of CRC cell growth
[150]	KDM5B	H3K4me3	Upregulated	Promoting Wnt/β-catenin signaling by reducing Caudal Type Homeobox 2 levels	Promoting CRC cell proliferation
[151]	KDM5B	H3K4me3	Upregulated	Inhibition of CC chemokine ligand 14 and Wnt/β-catenin activation	Promoting CRC cell proliferation and invasiveness
[152]	KDM5C	H3K4me3	NA	Downregulation of ATP Binding Cassette Subfamily C Member 1 expression	Promotion of multidrug resistance
[153]	KDM5C	H3K4me3	Upregulated	Inhibition of methyltransferase-like 14 transcription	Mediating CRC cell migration, invasion and metastasis
[154]	KDM5C	H3K4me3	Upregulated	Inhibition of prefoldin subunit 5 transcription to enhance the transcriptional activity of c-Myc	KDM5C knockdown inhibited malignant behavior of CRC cells
[155]	KDM5D	H3K4me3	Downregulated	Inhibition of FKBP Prolyl Isomerase 4 transcription by inhibiting E2F transcription factor 1 expression	KDM5D overexpression inhibited CRC growth and metastasis in vitro and in vivo
[28]	KDM5D	H3K4me2/3	Upregulated	Disruption of cancer cell adhesion properties and tumor immunity	Mediating sex differences in KRAS mutant CRC
[156]	KDM5D	NA	Downregulated	NA	Direct control of CRC cell proliferation, migration and invasion

Abbreviations: CRC, colorectal cancer; HDM, histone demethylase; NA, not available.

**Table 7 cimb-47-00267-t007:** A summary of research on the role of LSD family inhibitors in CRC.

References	HDM Inhibitors	Targeted HDM	Histone Demethylation Sites	Mechanisms of Action	Impact on CRC	IC50
[178]	Polyamine analogs	KDM1A	H3K4me2	Induction of re-expression of multiple abnormally silenced genes	NA	1 μM
[179]	Oligoamine analogs	KDM1A	H3K4me1/2	Induction of re-expression of multiple abnormally silenced genes	Inhibition of tumor growth in vitro and in vivo	5 μM
[180]	Compound **12**	KDM1A	H3K9me2	NA	Inhibition of HCT-116 cell proliferation in vitro	0.013 μM
[181]	L05	KDM1A	H3K4me1/2	NA	Inhibition of CRC cell migration	0.21 μM
[182]	PEGylated 17i nanoassemblies	KDM1A	NA	NA	Improved anti-tumor effects in vivo	0.065 μM
[88]	CBB1003	KDM1A	No changes in global methylation levels	Downregulation of LGR5 levels and inactivation of Wnt/β-catenin pathway	Inhibits CRC cell proliferation and colony formation	250.4 μM
[89]	ORY-1001 and GSK2879552	KDM1A	NA	NA	Reduced cell migration	NA
[96]	ZY0511	KDM1A	H3K4me1/2	Increased expression of the mTORC1 suppressor DNA Damage Inducible Transcript 4	Inhibition of CRC cell proliferation	1.7 nM
[97]	ZY0511	KDM1A	NA	Inhibition of Wnt/β-catenin signaling and DNA synthesis pathways	Inhibition of CRC cell proliferation in vitro and in vivo	1.4 nM
[98]	ZY0511	KDM1A	NA	Profoundly altered lipid metabolism in CRC cells	NA	1.7 nM
[95]	Tranylcypromine	KDM1A	H3K4me2	Suppression of EMT	Reduced migration and invasion of CRC cells	NA
[99]	Tranylcypromine hemisulfate	KDM1A	H3K9me2	Downregulation of tetraspanin 8 expression to inhibit EMT	Inhibition of CRC cell proliferation and migration	NA

Abbreviations: LSD, lysine-specific demethylase; CRC, colorectal cancer; HDM, histone demethylase; IC50, half maximal inhibitory concentrations; NA, not available; LGR5, Leucine Rich Repeat Containing G Protein-Coupled Receptor 5; EMT, epithelial–mesenchymal transition.

**Table 8 cimb-47-00267-t008:** A summary of research on the role of JMJD family inhibitors in CRC.

References	HDM Inhibitors	Targeted HDM	Histone Demethylation Sites	Mechanisms of Action	Impact on CRC	IC50
[107]	Daminozide	KDM2A	NA	Enhancement of nascent RNA transcription and 5-fluorouridine toxicity	Promotion of CRC cell death	NA
[183]	CBA-1	KDM3A	H3K9me2	Suppression of Wnt target genes	Inhibition of CRC cell proliferation in vitro	NA
[184]	IOX1	KDM3A/B	H3K9me2/3	Inhibition of Wnt target gene transcription	Inhibition of CRC tumorigenesis	NA
[185]	PROTACs using IOX1 as a warhead	KDM3A/B	NA	Inhibition of Wnt signaling	Elimination of colorectal CSCs and inhibition of tumor growth	8.9 μM
[186]	PKF118-310	KDM4A	H3K9me2/3	Impacts the cell cycle and induces apoptosis	Inhibition of CRC cell proliferation	10 μM
[187]	Benzo[b]tellurophenes (compound **1c**)	KDM4	H3K9me3	NA	Induction of colon cancer cell death	30.24 μM
[188]	QC6352	KDM4	H3K36me3	NA	Inhibition of CRC cell viability	13 nM
[140]	SD70; NSC636819	KDM4C; KDM4A/B	H3K9me3	Inhibition of ARID3B-induced target gene expression	Reversed CRC stem cell-like features	NA
[30]	Kaempferol	KDM4C	NA	Direct interaction with HNRNPK and HNRNPL to inhibit circ_0000345-mediated KDM4C/β-catenin signaling	Inhibits CRC cell migration	49.02/66.04 μM
[142]	Avenanthramide A	KDM4C	H3K9me3	Impairment of KDM4C/MIR17HG/GSK-3β negative feedback loop	Enhancing the therapeutic effect of 5-fluorouracil in CRC	NA
[145]	5-c-8HQ	KDM4D	H3K9me3	Inhibition of the Hedgehog signaling pathway	Inhibition of CRC cell proliferation and tumorigenesis	NA
[31]	5-c-8HQ	KDM4D	H3K9me3	Reduced PD-L1 expression by inhibiting interferon gamma receptor 1-STAT3-interferon regulatory factor 1 signaling	Enhancing the anti-tumor efficacy of PD-L1 antibodies	NA
[189]	24s	KDM4D	H3K9me3	NA	Inhibition of CRC cell proliferation and migration	0.023 μM
[190]	TACH101	KDM4	H3K36me3	NA	Inhibition of CRC cell proliferation and suppression of tumor growth in vivo	0.0004 (homogeneous time-resolved fluorescence)/0.085 (immunoblot) μM
[148]	HY-100014	KDM5A	H3K4me3	Impact on KDM5A enrichment at the TP73 promoter	NA	NA
[147]	CPI-455	KDM5A	H3K4me3	Promotion of cullin 4A expression	NA	10 nM
[153]	KDM5A-IN-1	KDM5C	H3K4me3	Promotion of methyltransferase-like 14 expression	NA	55 nM
[27]	GSK-J4	KDM6A/B	H3K27me3	Interruption of oxaliplatin-induced NOTCH signaling	Enhanced oxaliplatin-induced apoptosis in CRC	NA
[29]	GSK-J4	KDM6A/B	H3K27me3	Induction of global enhancer reprogramming	Attenuated malignant phenotype, increased chemosensitivity, and suppressed tumor-initiating cells and stemness-associated gene signatures	0.75–21.41 µM
[167]	GSK-J4	KDM6B	H3K27me3	Reduced C-X-C Motif Ligand 9 and 10	Reduced anti-tumor immunity	NA
[172]	Daminozide	KDM7B	H3K9me2 (and reduced H3K4me3)	Reduced expression of oncogenes KRAS, BRAF and c-Myc	Inhibition of CRC cell proliferation	NA

Abbreviations: CRC, colorectal cancer; HDM, histone demethylase; JMJD, Jumonji C domain-containing; IC50, half maximal inhibitory concentrations; NA, not available; PROTACs, Proteolysis Targeting Chimeras; ARID3B, AT-rich interaction domain 3B; PD-L1, Programmed cell death ligand 1.

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
