# Peer review of "Demystifying the Role of Histone Demethylases in Colorectal Cancer: Mechanisms and Therapeutic Opportunities"

_cimb, 2025, doi:10.3390/cimb47040267_

Round 1
Reviewer 1 Report
Comments and Suggestions for Authors
The review "Demystifying the Role of Histone Demethylases in Colorectal Cancer: Mechanisms and Therapeutic Opportunities" provides a detailed analysis of the role of histone demethylases in colorectal cancer. It is well-structured and offers valuable and insightful information. However, an introduction to the histone methylation mechanism is missing, which would help contextualize the topic before discussing the classification of histone demethylases. Additionally, including some figures would enhance readability and improve the overall comprehension of the review.
Author Response
Comment 1:The review "Demystifying the Role of Histone Demethylases in Colorectal Cancer: Mechanisms and Therapeutic Opportunities" provides a detailed analysis of the role of histone demethylases in colorectal cancer. It is well-structured and offers valuable and insightful information. However, an introduction to the histone methylation mechanism is missing, which would help contextualize the topic before discussing the classification of histone demethylases.
Response: Thank you for reviewing our manuscript and for your valuable comments. Your suggestions are very helpful for us to further improve the manuscript. We fully agree with you. In the revised version, we have briefly introduced the basic mechanism of histone demethylation before discussing the classification of histone demethylases. This will provide the necessary background knowledge for subsequent discussions on the classification and function of histone demethylases. We have added the following description:
"3. Overview of mechanisms of histone methylation, HDM classification and composition
In eukaryotic cells, DNA exists in the form of chromatin, and nucleosomes are the basic building blocks of chromatin. The core histones of the nucleosome (including histones H2A, H2B, H3, H4) have two structural domains: the histone folding domain and the amino-terminal domain. The amino-terminal domain is located outside the nucleosome core like a “histone tail” and is rich in amino acid residues that can be covalently modified[61]. Histone methylation is a key epigenetic modification that dynamically regulates gene expression by altering chromatin structure. This process involves the addition of methyl groups to lysine or arginine residues in histone tails and is mediated by histone methyltransferases (HMTs)[62]. Methylation can activate or repress transcription depending on the specific residue and degree of methylation (e.g., mono-, di-, or trimethylation.) HDMs promote the reversibility of these modifications, allowing cells to rapidly adapt to environmental and developmental cues[62]. HDMs remove methyl groups through an oxidation reaction-dependent catalytic mechanism and require specific cofactors (e.g., FAD and α-ketoglutarate) that play a key role in the catalytic process[63]. Histone demethylation affects the binding of transcription factors and other chromatin-associated proteins by altering the open or closed state of chromatin. "
Comment 2:Additionally, including some figures would enhance readability and improve the overall comprehension of the review.
Response: Your suggestions are very helpful for us to further improve the manuscript. We fully agree with you. We have added three additional figures, including Figure 1, Figure 2, and Figure 3. These figures will help to more clearly understand the classification, composition, and mechanism of action of HDM in CRC. Please see the revised manuscript.
Figure 1. Mechanistic Overview of HDMs in CRC.
Figure 2. HDM classification, composition, and substrate overview.
Figure 3. HDM inhibitors and their corresponding HDM targets in CRC.
Reviewer 2 Report
Comments and Suggestions for Authors
Overall Evaluation:
This manuscript provides a comprehensive overview of the role of histone demethylases (HDMs) in colorectal cancer (CRC) progression, with a particular focus on their mechanisms of action and therapeutic potential. However, the manuscript contains several issues that need to be addressed to improve its clarity, rigor, and overall quality for publication.
Specific Problems and Suggestions for Improvement:
- Lack of Clarity in the Introduction (Page 1, Lines 1-10):
Suggestion:Revise the introduction to clearly state the main research question, the significance of studying HDMs in CRC, and how this work contributes to advancing the field.
- Inconsistent Use of Terminology (Throughout the Document):
Suggestion:Maintain consistency in terminology throughout the manuscript. Choose one preferred term and use it consistently.
- Insufficient Explanation of HDM Classification and Function (Page 3, Lines 154-168):
Suggestion: Provide a more detailed and accessible explanation of HDM classification, their enzymatic functions, and their roles in epigenetic regulation.
- Lack of Comprehensive Citations (Throughout the Document):
Suggestion: Ensure that all claims are supported by appropriate citations. Review the literature and include references to the most relevant and recent studies.
- Overly Technical Language (Throughout the Document):
Suggestion: Define abbreviations when they are first used and limit the use of unnecessary abbreviations. Avoid overly technical language and provide explanations for complex concepts.
- Insufficient Discussion of Methodological Limitations (Page 20, Lines 19-21):
Suggestion: Include a more detailed discussion of the methodological limitations of the existing studies on HDMs in CRC, such as sample size, study design, and potential biases.
- Lack of Visual Aids (Throughout the Document):
Suggestion: Consider including figures, tables, or diagrams to visually represent complex mechanisms, pathways, or relationships discussed in the text. This will improve readability and comprehension.
- Inconsistent Formatting and Typographical Errors (Throughout the Document):
Suggestion: Carefully proofread the manuscript to correct any typographical errors and ensure consistent formatting throughout.
Specific Annotations:
Page 1, Line 6: Clarify the main research focus and significance of the study.
Page 3, Line 155: Provide a more detailed explanation of HDM classification and function.
Page 4, Line 94: Define "CRC" as "colorectal cancer" when first used.
Page 5, Line 196: Add a citation to support the claim that KDM1A is upregulated in CRC samples.
Page 10, Line 319: Provide more context on the role of 15-LOX-1 in CRC and how KDM3A regulates its expression.
Page 14, Line 123: Define abbreviations such as "GLUT1" and "LC3B" when first used.
Page 20, Line 19: Include a more comprehensive discussion of the methodological limitations of existing studies.
Throughout the Document: Proofread carefully to correct typographical errors and ensure consistent formatting.
Comments on the Quality of English Language
The English could be improved to more clearly express the research.
Author Response
Comment 1:Lack of Clarity in the Introduction (Page 1, Lines 1-10):
Suggestion:Revise the introduction to clearly state the main research question, the significance of studying HDMs in CRC, and how this work contributes to advancing the field.
Response:Thank you for your valuable comments on the manuscript. We have revised the introduction section according to your suggestions, and the specific adjustments in the Introduction are as follows:
“Key questions include how specific HDMs dynamically regulate histone methylation patterns to drive CRC malignancy, what are the upstream regulators and downstream effectors of HDMs in the CRC signaling pathway, and whether targeting HDMs can overcome drug resistance or immune evasion in CRC. HDMs are emerging as key epigenetic regulators of CRC that modulate tumor cell plasticity, stemness, and microenvironmental interactions. Importantly, HDM dysregulation is associated with chemotherapy resistance, making them promising therapeutic targets. This review provides a comprehensive overview of current knowledge on HDMs in CRC, including classification and composition of HDMs, elucidation of their roles in CRC hallmarks, dissection of the interactions between HDMs and signaling pathways, critical evaluation of preclinical HDM inhibitors, and future research directions.”
Comment2:Inconsistent Use of Terminology (Throughout the Document):
Suggestion:Maintain consistency in terminology throughout the manuscript. Choose one preferred term and use it consistently.
Response:We sincerely appreciate the reviewer’s valuable feedback regarding terminology consistency. We have carefully revised the manuscript to ensure uniform usage of key terms (e.g., LSD1 replaced by KDM1A).
Comment 3:Insufficient Explanation of HDM Classification and Function (Page 3, Lines 154-168):
Suggestion: Provide a more detailed and accessible explanation of HDM classification, their enzymatic functions, and their roles in epigenetic regulation.
Response:We strongly agree with your comments and have added a description of HDM classification, function, and mechanism of action to this section as follows:
“In eukaryotic cells, DNA exists in the form of chromatin, and nucleosomes are the basic building blocks of chromatin. The core histones of the nucleosome (including histones H2A, H2B, H3, H4) have two structural domains: the histone folding domain and the amino-terminal domain. The amino-terminal domain is located outside the nucleosome core like a “histone tail” and is rich in amino acid residues that can be covalently modified[61]. Histone methylation is a key epigenetic modification that dynamically regulates gene expression by altering chromatin structure. This process involves the addition of methyl groups to lysine or arginine residues in histone tails and is mediated by histone methyltransferases (HMTs)[62]. Methylation can activate or repress transcription depending on the specific residue and degree of methylation (e.g., mono-, di-, or trimethylation.) HDMs promote the reversibility of these modifications, allowing cells to rapidly adapt to environmental and developmental cues[62]. HDMs remove methyl groups through an oxidation reaction-dependent catalytic mechanism and require specific cofactors (e.g., flavin adenine dinucleotide and α-ketoglutarate) that play a key role in the catalytic process[63]. Histone demethylation affects the binding of transcription factors and other chromatin-associated proteins by altering the open or closed state of chromatin.”
“A schematic overview of the mechanism of HDMs in CRC was presented in Figure 1. The vast majority of HDMs are lysine demethylases (KDMs). KDMs are divided into two main groups based on their catalytic mechanisms and structural domains: the lysine-specific demethylase (LSD) family and the Jumonji C (JmjC) domain-containing (JMJD) family[64, 65]. The LSD family mainly removes mono- and dimethyl groups on lysine residues (me1 and me2), whereas the JMJD family removes mono-, di- and trimethyl groups (me1, me2, and me3) on lysine residues[17, 19]. The LSD family contains a flavin adenine dinucleotide-dependent amine oxidase structural domain and relies on flavin adenine dinucleotide to catalyze the demethylation of mono/demethylated lysines, whereas the JMJD family contains the JmjC structural domain and removes methylation modifications via Fe²⁺ and α-ketoglutarate-dependent oxidative reactions[66, 67].”
Comment 4: Lack of Comprehensive Citations (Throughout the Document):
Suggestion: Ensure that all claims are supported by appropriate citations. Review the literature and include references to the most relevant and recent studies.
Response: We sincerely appreciate the reviewer’s suggestion to strengthen the manuscript’s citations. After careful scrutiny, we ensured that we cited appropriate, adequate, and updated literature to reflect current knowledge and recent advances.
Comment 5:Overly Technical Language (Throughout the Document):
Suggestion: Define abbreviations when they are first used and limit the use of unnecessary abbreviations. Avoid overly technical language and provide explanations for complex concepts.
Response: We sincerely appreciate the reviewer’s valuable feedback regarding the accessibility of our manuscript. We have improved the readability of the article and the specification of the use of acronyms as requested, please see the revised manuscript.
Comment 6:Insufficient Discussion of Methodological Limitations (Page 20, Lines 19-21):
Suggestion: Include a more detailed discussion of the methodological limitations of the existing studies on HDMs in CRC, such as sample size, study design, and potential biases.
Response: We sincerely appreciate the reviewer’s insightful suggestion to critically evaluate the limitations of current HDM research in CRC. We have expanded the Discussion section to explicitly address key methodological challenges and biases. Below are the specific revisions in the 7. Conclusions and future directions section:
“ Some of the methodological limitations of the existing studies are worth noting. Many preclinical studies do not include clinical patient cohorts or use small patient xenograft cohorts or limited cell line models that may not adequately reflect the molecular diversity of CRC. Clinical relevance often relies on retrospective The Cancer Genome Atlas data with uneven racial/ethnic representation. Many HDM inhibitor-related studies have used monolayer cultures, neglecting the effect of the three-dimensional tumor microenvironment on drug response. ”
Comment 7:Lack of Visual Aids (Throughout the Document):
Suggestion: Consider including figures, tables, or diagrams to visually represent complex mechanisms, pathways, or relationships discussed in the text. This will improve readability and comprehension.
Response: We sincerely appreciate the reviewer’s constructive suggestion to enhance the manuscript’s clarity through visual aids. We have added three additional figures, including Figure 1, Figure 2, and Figure 3.These figures will help to more clearly understand the classification, composition, and mechanism of action of HDM in CRC. Please see the revised manuscript.
Figure 1. Mechanistic Overview of HDMs in CRC.
Figure 2. HDM classification, composition, and substrate overview.
Figure 3. HDM inhibitors and their corresponding HDM targets in CRC.
Comment 8:Inconsistent Formatting and Typographical Errors (Throughout the Document):
Suggestion: Carefully proofread the manuscript to correct any typographical errors and ensure consistent formatting throughout.
Response: We sincerely appreciate the reviewer’s meticulous attention to detail regarding formatting consistency and typographical accuracy. We have thoroughly revised the manuscript to address these issues. Please see the revised manuscript.
Specific Annotations:
1.Page 1, Line 6: Clarify the main research focus and significance of the study.
Response: We strongly agree with your comments and have added these sections to the last paragraph of the introduction, as follows:
“Key questions include how specific HDMs dynamically regulate histone methylation patterns to drive CRC malignancy, what are the upstream regulators and downstream effectors of HDMs in the CRC signaling pathway, and whether targeting HDMs can overcome drug resistance or immune evasion in CRC. HDMs are emerging as key epigenetic regulators of CRC that modulate tumor cell plasticity, stemness, and microenvironmental interactions. Importantly, HDM dysregulation is associated with chemotherapy resistance, making them promising therapeutic targets. This review provides a comprehensive overview of current knowledge on HDMs in CRC, including classification and composition of HDMs, elucidation of their roles in CRC hallmarks, dissection of the interactions between HDMs and signaling pathways, critical evaluation of preclinical HDM inhibitors, and future research directions.”
2.Page 3, Line 155: Provide a more detailed explanation of HDM classification and function.
Response: We strongly agree with your comments and have added these descriptions to this section as follows:
“3.Overview of mechanisms of histone methylation, HDM classification and composition
In eukaryotic cells, DNA exists in the form of chromatin, and nucleosomes are the basic building blocks of chromatin. The core histones of the nucleosome (including histones H2A, H2B, H3, H4) have two structural domains: the histone folding domain and the amino-terminal domain. The amino-terminal domain is located outside the nucleosome core like a “histone tail” and is rich in amino acid residues that can be covalently modified[61]. Histone methylation is a key epigenetic modification that dynamically regulates gene expression by altering chromatin structure. This process involves the addition of methyl groups to lysine or arginine residues in histone tails and is mediated by histone methyltransferases (HMTs)[62]. Methylation can activate or repress transcription depending on the specific residue and degree of methylation (e.g., mono-, di-, or trimethylation.) HDMs promote the reversibility of these modifications, allowing cells to rapidly adapt to environmental and developmental cues[62]. HDMs remove methyl groups through an oxidation reaction-dependent catalytic mechanism and require specific cofactors (e.g., flavin adenine dinucleotide and α-ketoglutarate) that play a key role in the catalytic process[63]. Histone demethylation affects the binding of transcription factors and other chromatin-associated proteins by altering the open or closed state of chromatin.
A schematic overview of the mechanism of HDMs in CRC was presented in Figure 1. The vast majority of HDMs are lysine demethylases (KDMs). KDMs are divided into two main groups based on their catalytic mechanisms and structural domains: the lysine-specific demethylase (LSD) family and the Jumonji C (JmjC) domain-containing (JMJD) family[64, 65]. The LSD family mainly removes mono- and dimethyl groups on lysine residues (me1 and me2), whereas the JMJD family removes mono-, di- and trimethyl groups (me1, me2, and me3) on lysine residues[17, 19]. The LSD family contains a flavin adenine dinucleotide-dependent amine oxidase structural domain and relies on flavin adenine dinucleotide to catalyze the demethylation of mono/demethylated lysines, whereas the JMJD family contains the JmjC structural domain and removes methylation modifications via Fe²⁺ and α-ketoglutarate-dependent oxidative reactions[66, 67]. ”
3.Page 4, Line 94: Define "CRC" as "colorectal cancer" when first used.
Response:We have noted the abbreviation “CRC” when the term “colorectal cancer” is first used.
4.Page 5, Line 196: Add a citation to support the claim that KDM1A is upregulated in CRC samples.
Response: We have added a relevant citation to this description as follows:"Hayami et al. first demonstrated that KDM1A was significantly upregulated in CRC samples from Japanese patients[81].”
81. Hayami S, Kelly JD, Cho HS, Yoshimatsu M, Unoki M, Tsunoda T, et al. Overexpression of LSD1 contributes to human carcinogenesis through chromatin regulation in various cancers. Int J Cancer. 2011;128(3):574-86. doi: 10.1002/ijc.25349.
5.Page 10, Line 319: Provide more context on the role of 15-LOX-1 in CRC and how KDM3A regulates its expression.
Response: We strongly agree with your comments and have added the following description to the corresponding section as suggested:
“Accumulating evidence suggests that 15-LOX-1 inhibits inflammation-driven colorectal tumorigenesis and suppresses signaling pathways that play a major role in chronic inflammation-promoting CRC, such as TNF-α, IL-1β/NF-κB, and IL-6/STAT3 pathways[110].”
“Knockdown of KDM3A in Caco-2 cells reduced KDM3A recruitment to the 15-LOX-1 promoter with a marked increase in H3K9me2 levels[111]. However, the specific molecular mechanism through which KDM3A regulates 15-LOX-1 levels remains unrevealed.”
6.Page 14, Line 123: Define abbreviations such as "GLUT1" and "LC3B" when first used.
Response: We have removed these unnecessary abbreviations based on your previous comments and instead show their definitions directly (glucose transporter 1 and light chain 3 beta).
7.Page 20, Line 19: Include a more comprehensive discussion of the methodological limitations of existing studies.
Response:We sincerely appreciate the reviewer’s insightful suggestion to critically evaluate the limitations of current HDM research in CRC. We have expanded the Discussion section to explicitly address key methodological challenges and biases. Below are the specific revisions in the 7. Conclusions and future directions section:
“ Some of the methodological limitations of the existing studies are worth noting. Many preclinical studies do not include clinical patient cohorts or use small patient xenograft cohorts or limited cell line models that may not adequately reflect the molecular diversity of CRC. Clinical relevance often relies on retrospective The Cancer Genome Atlas data with uneven racial/ethnic representation. Many HDM inhibitor-related studies have used monolayer cultures, neglecting the effect of the three-dimensional tumor microenvironment on drug response. ”
8.Throughout the Document: Proofread carefully to correct typographical errors and ensure consistent formatting.
Response: We sincerely appreciate the reviewer’s meticulous attention to detail regarding formatting consistency and typographical accuracy. We have thoroughly revised the manuscript to address these issues. Please see the revised manuscript.
Comments on the Quality of English Language
The English could be improved to more clearly express the research.
Response: We fully agree with your suggestions and have improved the presentation, grammar, etc. of the manuscript.
Reviewer 3 Report
Comments and Suggestions for Authors
In the manuscript, Liu et al. comprehensively reviewed the role of histone demethylases (HDMs) in colorectal cancer (CRC), including their mechanisms of action and therapeutic potential. This review provides a valuable resource and can support future research on HDM-based therapies in CRC. However, there are several areas where the manuscript could be improved:
- The authors included most key members from the LSD and JMJD families and provided detailed information on their functions in CRC. However, they rely heavily on text and tables. The authors should add several simple and informative figures to make the information more straightforward and reader-friendly. For example: 1. A summary figure showing HDMs and their target histone marks (e.g., H3K4me3, H3K9me2, H3K27me3); 2. Diagrams showing how HDMs regulate CRC progression through key signaling pathways such as Wnt/β-catenin, PI3K-AKT, and TGF-β; 3. A figure that groups HDMs with similar functions or mechanisms together; 4. A schematic showing HDM inhibitors and their targets in CRC.
- Although there are currently no approved clinical trials of HDM inhibitors specifically for CRC, some HDM inhibitors have entered clinical trials for other cancer types, such as leukemia and small cell lung cancer (10.1186/s13045-019-0811-9). The authors should include a summary of these trials. This would suggest the therapeutic potential of HDM inhibitors and provide references for developing HDM-based treatments in CRC.
Author Response
Comment 1:The authors included most key members from the LSD and JMJD families and provided detailed information on their functions in CRC. However, they rely heavily on text and tables. The authors should add several simple and informative figures to make the information more straightforward and reader-friendly. For example: 1. A summary figure showing HDMs and their target histone marks (e.g., H3K4me3, H3K9me2, H3K27me3); 2. Diagrams showing how HDMs regulate CRC progression through key signaling pathways such as Wnt/β-catenin, PI3K-AKT, and TGF-β; 3. A figure that groups HDMs with similar functions or mechanisms together; 4. A schematic showing HDM inhibitors and their targets in CRC.
Response:We strongly agree with your comments and have added three different figures to the manuscript as suggested, please see the revised manuscript.
Figure 1. Mechanistic Overview of HDMs in CRC.
Figure 2. HDM classification, composition, and substrate overview.
Figure 3. HDM inhibitors and their corresponding HDM targets in CRC.
Comment 2:Although there are currently no approved clinical trials of HDM inhibitors specifically for CRC, some HDM inhibitors have entered clinical trials for other cancer types, such as leukemia and small cell lung cancer (10.1186/s13045-019-0811-9). The authors should include a summary of these trials. This would suggest the therapeutic potential of HDM inhibitors and provide references for developing HDM-based treatments in CRC.
Response:We strongly agree with your comments and have summarized these HDM inhibitor-related clinical trials in the corresponding sections below:
“However, it is encouraging that clinical trials are already underway exploring the efficacy and safety of HDM inhibitors (mainly KDM1A) in other cancer types. A recent review provides a detailed summary of clinical trials of KDM1A inhibitors in other cancers, including small cell lung cancer, hematologic malignancies, acute myeloid leukemia, myelodysplastic syndromes, Ewing's sarcoma, solid tumors, myelofibrosis, essential thrombocythemia, prostate cancer, sarcomas, and ovarian-related tumors (please refer to this review for specific details and trial numbers) [191]. All clinical trials are in Phase I, I/II, or II[191]. However, there is still a lack of clinical trials conducted on the effects of JMJD family inhibitors in cancer.”
191. Cai W, Xiao C, Fan T, Deng Z, Wang D, Liu Y, et al. Targeting LSD1 in cancer: Molecular elucidation and recent advances. Cancer Lett. 2024;598:217093. doi: 10.1016/j.canlet.2024.217093.
Round 2
Reviewer 2 Report
Comments and Suggestions for Authors
The author has answered my questions very well, and I no longer have any questions.
Comments on the Quality of English LanguageThe English could be improved to more clearly express the research.
Reviewer 3 Report
Comments and Suggestions for Authors
Thanks for your responses to my comments. I agree that the manuscript can be accepted in its current format, thanks.